# ASMG: Data Structure-Aware Routing via Incremental Subspace Learning for MoE

## Abstract

Mixture-of-Experts (MoE) models scale model capacity efficiently by selectively routing inputs to a subset of specialized experts. However, their performance critically hinges on the gating mechanism, which is typically implemented as a shallow linear projection followed by a softmax or sigmoid activation. This minimal design lacks the representational capacity to capture structural variations in the input, often resulting in weak expert specialization and suboptimal routing. To address this limitation, we propose **A**daptive **S**tructure-Aware **M**oE **G**ating (**ASMG**), a data-driven gating mechanism that dynamically interpolates between a standard learnable gating matrix and an evolving principal subspace learned via the Generalized Hebbian Algorithm (GHA). By tracking input structure with iterative basis updates, ASMG enables the gating function to remain both task-supervised and structure-aware throughout training. We validate our method through (i) a highly controlled synthetic task based on multinomial HMMs and (ii) extensive real-world benchmarks spanning multiple domains and training regimes, including both finetuning and pretraining. Across a wide range of evaluations, ASMG achieves consistent gains over strong MoE baselines. Moreover, optionally enabling unsupervised GHA updates at test time further improves robustness under distribution shifts, offering an online adaptation mechanism that enhances standard gating with stronger OOD resilience.

## 1 Introduction

The Mixture-of-Experts (MoE) (Jacobs et al., 1991; Jordan & Jacobs, 1993), is a classic design that substantially scales up the model capacity with minimal computation overheads. Recently, incorporating MoE into the deep neural networks has achieved remarkable successes (Eigen et al., 2014; Shazeer et al., 2017). Among the various variants of MoE, a sparsely gated MoE, namely SMoE, efficiently scales up Transformers (Lepikhin et al., 2020; Fedus et al., 2022), resulting in significant accuracy enhancements. Consisting of multiple subnetworks (experts) that specialize in different tasks, MoE can benefit from a large pool of specialized knowledge by the selective input gating to a subset of these experts. This scalable and flexible nature of MoE is particularly appealing for Large Language Models (LLMs), which struggle with massive model sizes and diverse training datasets. Recent LLMs actively incorporate MoE layers (Jiang et al., 2024; Dai et al., 2024).

The core component that fundamentally governs the performance of MoE is the gating mechanism, which routes each input to a subset of experts based on its relevance. Despite its central role, the current routing design remains overly simplistic, typically a shallow linear projection followed by softmax or sigmoid activation (Fedus et al., 2022; Jiang et al., 2024; Aghdam et al., 2024; Dai et al., 2024). Such a minimal design has limited capacity to capture the structural diversity of real input distributions, often leading to weak expert specialization. To compensate for this, many prior efforts concentrate on mitigating load balancing issue, a tendency of sparse MoE routers to over-utilize a small subset of experts, primarily through adding auxiliary regularizers such as the load-balancing loss (Shazeer et al., 2017) and the GShard loss (Lepikhin et al., 2020), or alternative routing paradigms such as expert-choice routing (Zhou et al., 2022; Wang et al., 2024a).

However, heuristically enforcing load balance through explicit regularization is a result-oriented constraint that overlooks the underlying issue: the router lacks awareness of the input distribution and fails to distinguish the core structural variations necessary for meaningful routing. This pulls the

gating away from the central purpose of MoE, which is assigning heterogeneous inputs to specialized sub-networks, the experts, so that fully leverages the extremely wide networks while keeping the computation modest. Indeed, recent studies point out that using explicit balancing loss can overly constrain the routing decision, degrading model performance (Wang et al., 2024b; Qiu et al., 2025).

The principal role of the gating mechanism is not to balance load, but to be data-aware and capable of discerning and responding to input variations so as to promote input-expert specialization. Motivated by this, we analyze the limited representational capacity of the conventional gating function and develop a novel balancing-free and input-aware method to augment linear routing with data-driven basis structure learned in an unsupervised way.

**Synthetic Experimental Setup**  For better principled investigation of routing behavior across various data structures, we carefully construct a synthetic setup based on sequential modeling by multinomial Hidden Markov Models (HMMs) for next token prediction task. Following the settings used in GINC dataset generation (Xie et al., 2022), we simulate a language modeling scenario using sequences generated from a multinomial HMM, where latent entity-property structures govern token transitions. Each entity-property pair constructs a distinct hidden state and the transitions between the hidden states are modeled by a mixture of HMMs. This setup provides inputs with clear, interpretable structural variations—entity, property transitions, and mixed contextual dynamics—while remaining analytically tractable.

Within this controlled framework, we study a basis-guided gating mechanism grounded in unsupervised representation learning. At the core of our method lies the Generalized Hebbian Algorithm (GHA), an iterative algorithm that incrementally learns the principal components of the input distribution. The learned orthonormal basis vectors capture the dominant directions of input variance, providing a compact and semantically meaningful representation of the data structure. By constructing a latent routing space through learnable linear combinations of these basis vectors, the gating becomes data structure-aware, facilitating a better input-expert specialization.

Our contributions and outline for this paper are summarized as follows:

- We propose ASMG, a basis-guided MoE gating mechanism using the principal subspace learning via Generalized Hebbian Algorithm (GHA), and introduce a latent mixing scheme that naturally promotes balanced and structure-aware routing.
- We construct a controlled synthetic testbed based on multinomial HMMs, enabling principled analysis of routing behavior under structured sequence variations.
- We evaluate ASMG across a wide range of real experimental setups.
  (i) Zero-shot evaluation on pretrained LLM-MoE 182M model.
  (ii) Finetuning on large (Qwen1.5-MoE 14B) and moderate-sized (BERT-Large) language models across various benchmarks.
  (iii) Test-time adaptation for OOD generalization in both language and vision domain.
- Across synthetic, language, and vision experiments, we show that ASMG consistently improves expert specialization, routing quality, and downstream task performance, and further supports test-time adaptation that enhances OOD robustness.

## 2 BACKGROUND

**Mixture of Experts**  In Transformer-based models, a Mixture-of-Experts (MoE) layer, which consists of multiple expert networks $\{f_1, ..., f_K\}$ and a gating network $\mathbf{G}$, i.e., router, replaces the feed-forward network (FFN) after the self-attention. The router, typically implemented as a shallow linear-softmax network, determines which subset of experts should process each input. Based on the design choice of gating functions, the MoE can be classified into two types of categories; dense MoE that activates all experts weighted by computed gating coefficients (Pan et al., 2024; Dou et al., 2024) and sparse MoE (SMoE) that sparsely activates a few subset of experts, typically with top-k mechanism (Fedus et al., 2022; Dai et al., 2024). To efficiently scale up the model, SMoE has been preferred over dense variants in many domains (Cai et al., 2025; Li et al., 2023). Throughout the paper, we will also focus on improving the sparse gating.

**Generalized Hebbian Algorithm**  As a generalization of Oja's rule (Oja & Karhunen, 1985), which provides an incremental solution for estimating the first eigenvector of the data covariance, i.e., the

first principal component of PCA, the Generalized Hebbian Algorithm (GHA) (Sanger, 1989), an unsupervised learning algorithm, incrementally estimates the top-$K$ principal components of a data distribution. Given input vectors $\mathbf{x} \in \mathbb{R}^d$, GHA learns an orthonormal basis matrix $\mathbf{V} \in \mathbb{R}^{K \times d}$ such that each row $\mathbf{v}_i$ approximates the $i$-th principal direction of the input covariance matrix. At each iteration, GHA performs the following update for each component $i \in \{1, \dots, K\}$:

$$\mathbf{v}_i \leftarrow \mathbf{v}_i + \eta \left( y_i \mathbf{x} - y_i \sum_{j=1}^{i} y_j \mathbf{v}_j \right), \text{ where } y_i = \mathbf{v}_i^\top \mathbf{x}$$

Here, $\eta$ is the learning rate controlling the step size of the incremental GHA updates. Iterative updates of $\mathbf{v}_i$ following this rule enforces orthogonality among the learned components while aligning them with the dominant directions of variance in the input space. Unlike standard PCA, GHA operates in an online fashion and does not require explicit computation of the covariance matrix, making it suitable for streaming or mini-batch scenarios. Since PCA components correspond to the right singular vectors of the data matrix $\mathbf{X}$ up to scale (i.e., the matrix $V$ in the singular value decomposition (SVD) of $\mathbf{X} = U\Sigma V^\top$), GHA can be seen as an online approximation of the top $K$ right singular vectors of $\mathbf{X}$. Thus, it provides a principal basis directly from the streaming input.

**Multinomial Hidden Markov Model**   A Hidden Markov Model (HMM) (Baum & Petrie, 1966) is a generative probabilistic model for sequential data where each observation is emitted by a latent state, and these latent states evolve by a Markov process where each hidden state depends only on the previous state and emits an observation independently by emission probabilities. Formally, let $\{h_t\}_{t=1}^T$ be the hidden state sequence and $\{o_t\}_{t=1}^T$ the observations. The generative process is defined by an initial distribution $\pi = p(h_1)$, transition probabilities $\mathcal{H} = p(h_t|h_{t-1})$, and emission probabilities $\mathcal{E} = p(o_t|h_t)$. The joint probability of an entire sequence of hidden states and observations, parameterized by $\theta = \{\pi, \mathcal{H}, \mathcal{E}\}$, is given by:

$$p(o_{1:T}, h_{1:T} \mid \theta) = \pi(h_1) \prod_{t=2}^{T} p(h_t \mid h_{t-1}) \prod_{t=1}^{T} p(o_t \mid h_t)$$

A multinomial HMM refers to a HMM where the emission probabilities are modeled by a categorical (multinomial) distribution over a finite set of discrete symbols. Each hidden state $h_t \in \{1, \dots, N\}$ is associated with a probability distribution over a finite set of observations $\mathcal{O} = \{1, \dots, M\}$. The emission probabilities are defined by an emission matrix $\mathcal{E} \in \mathbb{R}^{N \times M}$, with each row specifying the likelihood of emitting each symbol given a state. The model follows the standard HMM structure with an initial state distribution $\pi \in \mathbb{R}^N$ and a transition matrix $\mathcal{H} \in \mathbb{R}^{N \times N}$. This particular model is well suited for sequential data consisting of discrete symbols, such as natural language.

## 3   MAIN RESULTS ON SYNTHETIC EXPERIMENTS

### 3.1   STRUCTURE OF MOE LAYER

**MoE Network**   We define the architecture of the Mixture-of-Experts (MoE) network, used as a baseline MoE layer in our synthetic experiments. The model consists of two main components: a gating module $\mathbf{G}$ and a set of experts $\{f_k\}_{k=1}^K$ where $\mathbf{W}_k \in \mathbb{R}^{d' \times d}$ is the weight matrix of expert $k$. For expert selection, we adopt a top-$k$ selection mechanism throughout the paper, where $k$ is a predefined hyperparameter. Given the input $\mathbf{x} \in \mathbb{R}^d$, the routing score for expert $k$ is computed as $s_k = \text{Softmax}_k(\mathbf{x}^\top \mathbf{g}_k)$, where $\mathbf{g}_k \in \mathbb{R}^d$ is the $k$-th gating vector. The final output $\hat{\mathbf{y}} \in \mathbb{R}^{d'}$ of the MoE is computed as $\hat{\mathbf{y}} = \sum_{k \in \text{Topk}(\{s_j\}_{j=1}^K, k)} s_k \cdot f_k(\mathbf{x})$ where $f_k(\mathbf{x}) = \mathbf{W}_k \mathbf{x}$.

**Gating Module**   Choosing the right gating vectors for $K$ experts determines how well each input is routed to a subset of experts based on its semantic relevance. We consider two types of gating modules depending on how to derive the gating vectors:

- **Vanilla Gating Module**: This corresponds to the standard linear projection where $\mathbf{g}_k$ is a learnable gating vector trained end-to-end with the rest of the model. We denote the collection of these gating vectors as a learnable matrix $\mathbf{W}_g \in \mathbb{R}^{K \times d}$.

- **GHA-based Gating Module**: Here, we replace the learnable gating vectors using a set of principal basis vectors derived from the GHA. Let $\mathbf{V} \in \mathbb{R}^{K \times d}$ be the orthonormal matrix where each row $\mathbf{v}_i$ approximates the $i$-th principal direction of the input distribution. To avoid assigning rigid hierarchical roles to individual principal basis and ensure that each gating vector lies in a semantically equivalent position within the routing space, we construct a latent routing subspace by linearly mixing these components with coefficient matrix $\mathbf{R} \in \mathbb{R}^{K \times K}$ as $\mathbf{Z} = \mathbf{RV}$, $\mathbf{z}_k = \mathbf{r}_k \mathbf{V} \in \mathbb{R}^d$ where $\mathbf{r}_k \in \mathbb{R}^K$ is $k$-th row of $\mathbf{R}$. The $k$-th row $\mathbf{z}_k \in \mathbb{R}^d$ of $\mathbf{Z}$ serves as the gating vector for the $k$-th expert. This creates a latent gating basis that spans the same routing subspace while avoiding direct alignment between raw principal components and specific experts.

## 3.2 Multinomial HMM for Language Modeling

To evaluate the generality of our method beyond simple classification, we construct a second synthetic task based on a structured language modeling scenario using Multinomial HMMs, following the GINC dataset generation procedure (Xie et al., 2022). This task is designed to assess the behavior of MoE routing under sequential input conditions, where tokens are generated from latent entity-property states, temporally evolving by mixtures of HMMs carrying distinct contexts.

**3.3.1 Data Generation** Each hidden state is a tuple $h_t = (v_t, s_t)$, where $v_t \in \mathcal{V}$ is an entity index and $s_t \in \mathcal{S}$ a property type. A global memory matrix $M \in \mathcal{O}^{|\mathcal{V}| \times |\mathcal{S}|}$ maps each entity–property pair deterministically to a token in the vocabulary $\mathcal{O}$. The transitions are defined as:

$$s_{t+1} \sim P(s_{t+1} \mid s_t; \theta), \quad \theta \sim p(\theta),$$
$$v_{t+1} \sim 0.9 \cdot \delta(v_{t+1} = v_t) + 0.1 \cdot I(v_{t+1} \mid v_t),$$

where $\theta \in \Theta$ is a concept parameter sampled from a uniform mixture of HMMs, each governing distinct property transition pattern. The $\delta(v_{t+1} = v_t)$ enforces entity persistence, yielding a sticky Markov process that keeps the same entity with probability 0.9 and switches uniformly with probability 0.1. At each step the emitted token is deterministic, $o_t = M[v_t, s_t]$ with $p(o_t \mid v_t, s_t) = 1$. Let $T$ be the sequence length, $n$ the number of training samples, and $t'$ the subsequence length. Each sample $(x_i, y_i)$ consists of input $x_i = [o_{t_i}, \ldots, o_{t_i + t' - 1}]$ and target $y_i = o_{t_i + t'}$. Thus, each sample uses $t' + 1$ tokens, giving total length $T = n(t' + 1) + t'$. The full sequence is $[x_1, y_1, o_{\text{delim}}, \ldots, x_n, y_n, o_{\text{delim}}, x_{\text{test}}]$, where $x_{\text{test}}$ is used to evaluate prediction of $y_{\text{test}} = o_T$.

**3.3.2 Model** For the language modeling task, we adopt a single-layer Transformer architecture whose final representation is augmented with a Mixture-of-Experts (MoE) module. The model consists of a token embedding layer, sinusoidal positional encoding, and a Transformer encoder block. The MoE module is placed as the feed-forward network (FFN) head, replacing the standard MLP. It consists of $K$ expert networks, each implemented as a single linear layer followed by a `SiLU` activation, mapping from $\mathbb{R}^d$ to $\mathbb{R}^{d'}$. A gating network computes token-wise routing scores over experts, and the top-$k$ experts are selected for each token. A shared decoder layer maps the output back to the token vocabulary. The model is trained with a standard language modeling objective using the cross-entropy loss.

**3.3.3 Basis Construction** In current experimental setup, the gating network receives hidden representations whose covariance structure naturally shifts as training progresses. Since the principal components captured by GHA typically characterize the dominant axes of variation in these input representations, updating them incrementally allows the router to remain aligned with the evolving input geometry. To achieve this, we apply m iterative GHA updates at each forward pass, continually refining the principal directions in response to the current hidden features. This incremental adaptation provides a stable, structure-aware routing subspace that tracks the underlying input dynamics, enabling more reliable expert–input specialization.

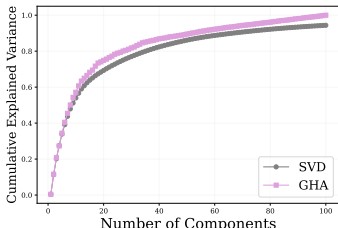

Figure 1: Cumulative explained variance comparison between SVD and GHA.

The iteration number $m$, as a tunable hyperparameter, controls the trade-off between approximation quality and computation cost. To assess the quality of the GHA-based basis, we compare it against SVD on hidden representations extracted from the Transformer

encoder, as shown in Fig. 1. We measure the cumulative explained variance of the top 100 components using $m = 3$ iterations per GHA update. GHA closely tracks the SVD, demonstrating that GHA effectively approximates the principal subspace of the input distribution in an incremental fashion. This supports the use of GHA as a online alternative to full SVD for basis construction. Detailed analysis of approximation quality with varying $m$ is in Section D

**3.3.4 Proposed Method** To maintain router aligned with the dynamically evolving input structure, we propose **A**daptive **S**tructure-**A**ware **M**oE **G**ating (**ASMG**), a strategy that interpolates between a standard learnable gating matrix and a principal gating subspace constructed via training-time GHA updates.

Given an input feature vector $\mathbf{x} \in \mathbb{R}^d$, ASMG computes two sets of routing logits, $l_{\text{vanilla}} = \mathbf{W}_g\mathbf{x}$ and $l_{\text{GHA}} = \mathbf{Z}\mathbf{x}$, where $\mathbf{W}_g \in \mathbb{R}^{K \times d}$ is the trainable gating matrix and $\mathbf{Z} = \mathbf{RV} \in \mathbb{R}^{K \times d}$ is the GHA-driven matrix formed by mixing the principal components $\mathbf{V} \in \mathbb{R}^{K \times d}$ with a learnable mixing matrix $\mathbf{R} \in \mathbb{R}^{K \times K}$. The final routing score $s \in \mathbb{R}^K$ is obtained through a learnable interpolation:

$$s = \text{Softmax}(\sigma(\alpha) \odot l_{\text{vanilla}} + (1 - \sigma(\alpha)) \odot l_{\text{GHA}}),$$

where $\alpha \in \mathbb{R}^K$ is a trainable vector of interpolation coefficients and $\sigma(\cdot)$ denotes the elementwise sigmoid activation. This interpolation allows ASMG to blend task-driven signals with GHA-based subspace learning, which ensures that routing remains both responsive to task objective supervision and aligned with the evolving input geometry. Overall, ASMG provides a unified routing framework that augments the conventional routing scheme with input-structure awareness. The full algorithmic details are summarized in Appendix B.

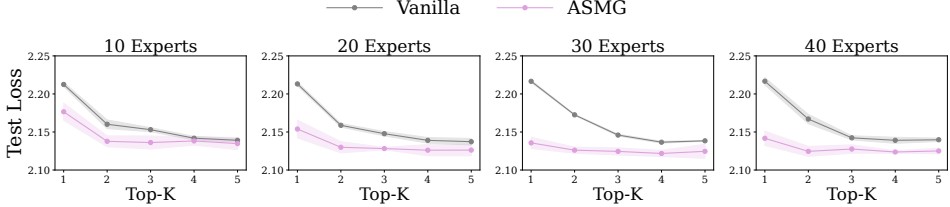

Figure 2: **Comparison of Test Loss.** We evaluate the performance of Vanilla and GHA-based gating methods under varying numbers of experts and top-$k$. Shaded regions indicate standard deviation over three random seeds.

**3.3.5 Observations** We configure the synthetic language modeling task with a vocabulary size of $|\mathcal{O}| = 50$, $|\mathcal{V}| = 15$ entities, and $|\mathcal{S}| = 15$ properties. The property transition is controlled by HMMs sampled from a mixture of distinct HMMs with $|\Theta| = 5$. After training the model, we measure the test loss across three random seeds to evaluate our proposed method against the standard Vanilla gating baseline. Figure 2 presents the performance comparison across varying numbers of experts, $K \in \{10, 20, 30, 40\}$, and different top-$k$ selections, $k \in \{1, 2, 3, 4, 5\}$.

ASMG consistently achieves lower test loss compared to Vanilla gating under all settings. As a dynamic interpolation between standard gating and structure-aware gating with an evolving basis updated via GHA, ASMG enables the gating to adaptively benefit from two pathways. First it leverages the flexibility of gradient-based optimization early in training. Second, ASMG can increasingly exploit the structural information captured by the GHA-driven basis as the representation stabilize with training. As a result, ASMG achieves superior specialization and expert routing even under dynamically shifting input distributions.

To investigate how the interpolation balance evolves between the learnable gating and the GHA-driven structure-aware routing, we track the mean interpolation coefficients $\sigma(\alpha)$ during training. Recall that $\alpha$ controls the weighting between the learnable logits $l_{\text{vanilla}}$ and the GHA-based logits $l_{\text{GHA}}$. Figure 3 shows the

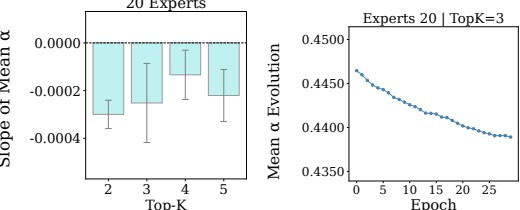

Figure 3: **Evolution of Interpolation Coefficient $\alpha$.** Left: Average slope of the mean $\sigma(\alpha)$ across different Top-$k$ settings. Right: Temporal evolution of mean $\sigma(\alpha)$ throughout training epochs.

slope of mean $\sigma(\alpha)$ across different top-$k$ settings and its trajectory over epochs. We observe a consistent decrease in $\sigma(\alpha)$ throughout training, indicating a uniform shift toward greater reliance on the GHA-driven gating. This trend reflects the stabilization of hidden representations, which enables the model to increasingly favor structure-aware subspace gating over gradient-driven gating across all expert sizes and top-$k$ settings. Although the absolute scale of $y$-axis values is small, the downward trend of $\sigma(\alpha)$ is consistent and clearly observed across all MoE configurations, confirming the robustness of this adaptation behavior. See Appendix G for full results.

**Specialization and Load Balance: The Two Pillars of MoE Routing**    Further, to understand whether the router becomes genuinely aware of the underlying input structure, and whether such input-awareness translates into stronger input–expert specialization, we examine the relationship between routing decisions and the latent factors of the generated data. In our synthetic multinomial HMM setup, the latent property $s_t$ is the primary source of token-level variation, making it the most informative target for assessing whether the gating mechanism responds meaningfully to changes in the input. Let $e_t \in \{1, \ldots, K\}$ denote the expert selected for token $x_t$ under top-1 routing scenario. We quantify structure-aligned specialization by computing the mutual information between expert assignments and latent properties,

$$I(e; s) \;=\; \sum_{e=1}^{K} \sum_{s \in \mathcal{S}} p(e, s) \, \log \frac{p(e, s)}{p(e)\, p(s)}, \tag{1}$$

$p(e, s)$ is the empirical joint distribution of expert–property pairs and $p(e)$ and $p(s)$ are its marginals. Higher values of $I(e; s)$ indicate that experts specialize along the true variation modes of the data rather than fragmenting arbitrarily. This metric therefore provides a direct measure of how effectively the routing mechanism captures meaningful structure in the input distribution.

In parallel, we evaluate how well each gating mechanism balances expert usage, since MoE performance also depends on preventing expert collapse and effectively distributing traffic across the expert pool. Let $p(e)$ denote the empirical fraction of tokens routed to expert $e$. We quantify load-uniformity using the normalized entropy

$$H_{\mathrm{norm}} \;=\; -\frac{\sum_{e=1}^{K} p(e) \, \log p(e)}{\log K}, \tag{2}$$

which lies in $[0, 1]$ and attains 1 when all experts are used uniformly. These two complementary metrics allows us to investigate how experts are being used (specialization via $I(e; s)$) from how evenly they are being used (capacity utilization via $H_{\mathrm{norm}}$), providing a more holistic view of routing behavior as the expert pool scales.

Figure 4 reports both quantities across different $K$, enabling a direct comparison of routing specialization and load balance between ASMG and the baseline gating module explicitly equipped with load balance regularizer (Shazeer et al., 2017) (denoted as Vanilla + LB). In the top panel, ASMG consistently achieves higher mutual information $I(e; s)$, with the gap widening as $K$ increases: Vanilla + LB improves up to $K = 30$ but collapses at $K = 40$ ($I(e; s) = 0.24$), whereas ASMG continues to rise, reaching 0.98. This indicates that ASMG benefits from increased routing capacity more reliably and maintains stable specialization even at larger expert sizes.

The bottom panel shows the normalized load entropy $H_{\mathrm{norm}}$. Both methods are reasonably balanced for smaller expert sizes, but Vanilla + LB becomes increasingly imbalanced as $K$ grows, with $H_{\mathrm{norm}}$ falling sharply from 0.43 at $K = 30$ to 0.17 at $K = 40$. In contrast, ASMG maintains stable and high load balance

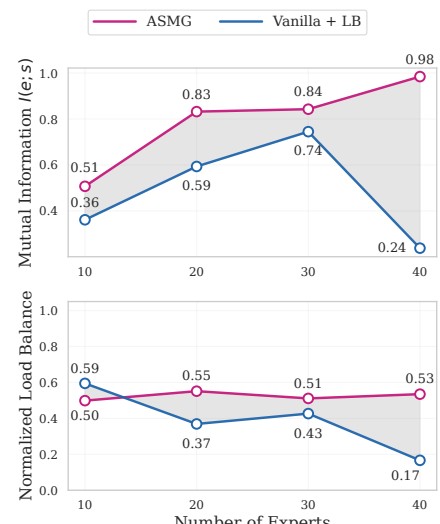

Figure 4: **Routing Specialization and Load Balance Across $K$.** ASMG shows stronger structure-aware specialization and more stable load balance compared to Vanilla + LB).

across all configurations. Together, these results show that ASMG achieves stronger structure-aware specialization without sacrificing load balance, whereas Vanilla + LB struggles to maintain either property at large size of expert pool.

Table 1: Training curve of ASMG and zero-shot scores of different routing methods. Results[1] from Wang et al. (2025). Best result colored in purple and second-best colored in green. ($m = 1$)

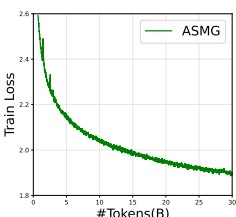

| Model | ARC-c | ARC-e | BoolQ | HellaSwag | LAMBADA | PIQA | RACE | Avg. |
|---|---|---|---|---|---|---|---|---|
| Hash[1] | 19.28 | 45.45 | 54.95 | 29.68 | 31.44 | 63.06 | 27.66 | 38.79 |
| Lory[1] | 20.31 | 42.97 | 49.54 | 28.75 | 32.35 | 62.24 | 27.75 | 37.70 |
| SparseMixer-v2[1] | 19.80 | 46.72 | 45.96 | 30.24 | 34.12 | 62.89 | 29.00 | 38.39 |
| EC[1] | 18.86 | 42.97 | 60.21 | 29.14 | 29.26 | 61.92 | 27.37 | 38.53 |
| dMoE[1] | 20.05 | 45.16 | 57.83 | 29.83 | 32.97 | 63.55 | 28.33 | 39.67 |
| ReMoE[1] | 20.22 | 46.68 | 54.16 | 30.26 | 35.94 | 63.55 | 29.38 | 40.03 |
| **ASMG** | 20.06 | 48.86 | 59.66 | 30.43 | 33.27 | 64.69 | 28.52 | 40.78 |

## 4    MAIN RESULTS ON REAL-DATA EXPERIMENTS

To validate the effectiveness of our proposed method ASMG, we evaluate ASMG across a broad spectrum of real-world settings spanning both language and vision domain.

### 4.1    LANGUAGE DOMAIN

#### 4.1.1    ZERO-SHOT EVALAUTION ON PRETRAINED LLM-MoE

**Training and Evaluation.**    To assess whether ASMG improves large-scale autoregressive modeling, we adopt a LLaMA-MoE backbone with 182M active parameters out of 777M total parameters. The model adopts a standard LLaMA-style decoder (GQA, SwiGLU, RoPE, RMSNorm), and each feed-forward block is replaced by an MoE layer with $E=8$ experts under Top-1 routing, matching the compute budget of the corresponding dense model. We follow the pretraining configuration of ReMoE (Wang et al., 2025), a recent MoE framework that replaces Top-$K$ gating with ReLU routing. All models are trained from scratch on The Pile for 30B tokens with sequence length 1024 and batch size 512. Top-$K$ baselines use the conventional load-balancing loss and include Hash (Roller et al., 2021), dMoE (Gale et al., 2022), Expert-Choice (EC) routing (Zhou et al., 2022), Lory (Zhong et al., 2024), and SparseMixer-v2 (Liu et al., 2024), whereas ASMG is trained without load balancing regularization. Here $m$ for ASMG is set to be 1 for efficient training. Following pretraining, we evaluate all models in a purely zero-shot manner on standard commonsense and reading-comprehension tasks: ARC-c, ARC-e, BoolQ, HellaSwag, LAMBADA, PIQA, and RACE.

**Results.**    Table 1 reports zero-shot accuracies across downstream tasks. ASMG attains the highest average score among all routing methods, improving over strong baselines such as SparseMixer-v2, EC, dMoE, and ReMoE. It also achieves the best performance on several individual tasks (e.g., BOOLQ, PIQA). These results indicate that integrating structure-aware gating into the routing mechanism can promote more stable and effective expert specialization during large-scale pretraining.

Table 2: LoRA-based finetuning performance on Qwen1.5-MoE (14B) across downstream tasks.

| Backbone | Method | GSM | MBPP | HE | Intent | Law | Summary | Translation | Avg |
|---|---|---|---|---|---|---|---|---|---|
| **Qwen1.5-MoE (14B)** | Base Model | 36.77 | 38.40 | 33.54 | 16.83 | 13.90 | 21.40 | 14.26 | 25.01 |
| | Vanilla + LB | 42.60 | 33.61 | 34.75 | 65.60 | 25.70 | 30.70 | 29.36 | 33.09 |
| | **ASMG** | 42.16 | 35.48 | 39.02 | 68.40 | 27.00 | 35.90 | 29.60 | 39.65 |

### 4.2    FINETUNING FOR LARGE-SCALE LLM-MoE

**Training and Evaluation.**    Following the same experimental setup from (Wang et al., 2024c), we next evaluate ASMG in a finetuning setting using the 14B Qwen1.5-MoE backbone. In detail, the router is fully updated for both Vanilla and ASMG, and all remaining modules, including the experts, are finetuned through LoRA following the standard LoRA-based post-training setup. We train on a mixed collection of downstream tasks, including GSM8K (Cobbe et al., 2021), CodeAlpaca (Chaudhary, 2023), and intent and law classification, summarization, and translation datasets (Wang et al.,

2024c). Evaluation is conducted on GSM8K, MBPP (Austin et al., 2021),HumanEval (Chen et al., 2021), as well as the corresponding held-out splits of the intent, law, summarization, and translation tasks. Under this setup, we compare the Top-k sparse routing Vanilla MoE model augmented with load-balancing regularization (Vanilla + LB) against ASMG under identical finetuning conditions.

**Results** As shown in Table 2, ASMG achieves higher average accuracy and yields consistent improvements on most tasks, demonstrating its effectiveness as a drop-in replacement for conventional routing during parameter-efficient adaptation.

Table 3: Performance comparison on 5 language tasks on GLUE benchmark. [1] from Guo et al. (2025)

| Algorithms (K, k) | CoLA | MRPC | QNLI | MNLI | RTE | Average |
|---|---|---|---|---|---|---|
| MoE (8,1)[1] | 64.10±0.94 | 90.14±0.60 | 92.48±0.21 | 86.56±0.06 | 73.04±2.13 | 81.26 |
| MoE (8,2)[1] | 64.51±0.81 | 90.19±0.17 | 92.39±0.08 | 86.70±0.23 | 74.85±1.96 | 81.73 |
| MoE (8,4)[1] | 64.94±0.62 | 89.74±0.99 | 92.52±0.12 | 86.57±0.28 | 75.09±1.84 | 81.77 |
| DynMoE (9, 7.1)[1] | 65.17±0.26 | 90.64±0.26 | 92.59±0.08 | 86.37±0.13 | 73.41±1.96 | 81.64 |
| **ASMG** (8,1) | 65.54±0.47 | 90.25±0.55 | 92.56±0.23 | 86.52±0.18 | 74.61±0.74 | 81.90 |
| **ASMG** (8,2) | 65.81±0.80 | 90.03±0.32 | 92.52±0.15 | 86.63±0.08 | 75.57±0.61 | 82.11 |
| **ASMG** (8,4) | 66.62±0.65 | 89.68±0.20 | 92.61±0.10 | 86.74±0.11 | 75.57±0.68 | 82.24 |

### 4.3 FINETUNING FOR MoE-AUGMENTED BERT ON GLUE

**Training and Evaluation** In this experiment, we follow the MoEfication setup (Zhang et al., 2022; Qiu et al., 2024) and finetune BERT-large (Devlin et al., 2019) models on the GLUE benchmark (Wang et al., 2019). Specifically, we evaluate on CoLA Warstadt et al. (2019), QNLI Wang et al. (2019), RTE Bentivogli et al. (2009), MNLI Xu et al. (2020), and MRPC Dolan & Brockett (2005) under multiple MoE settings with varying $K$ and top-$k$ selections. We compare our ASMG with other MoE variants, MoE with cosine routers (Li et al., 2023) and DynMoE (Guo et al., 2025). All reported results are averaged over three random seeds.

**Results** Table 3 shows that ASMG achieves consistent improvements in average accuracy across all $(K, k)$ settings, with the best result obtained at $(8, 4)$ reaching 82.24%, compared to 81.77% for the best MoE baseline and 81.64% for DynMoE.

To understand where these improvements originate, we examine two key desired properties for MoE routing under the current GLUE experimental setup. First, for expert–input specialization, we compute the $\ell_2$ distance between token embeddings routed to different experts (inter-expert) under the Top-2 selection setting. Distances are measured during the Early (first 100 steps) and Late (last 100 steps) phases of training, and normalized per phase and method to ensure scale-invariant comparison. The magnitude of change between the Early → Late phase, denoted as $\Delta$, reflects how effectively the router sharpens expert boundaries, an increase in inter-expert distance indicates clearer functional separation across experts.

As shown in Figure 5 (a), ASMG exhibits the largest improvement ($\Delta = +2.0\%$), whereas Vanilla and Vanilla+LB show minimal ($\Delta = +0.0\%$) or weaker ($\Delta = +1.4\%$) separation. These results indicate that ASMG develops more distinct and specialized expert roles compared to conventional routing mechanisms.

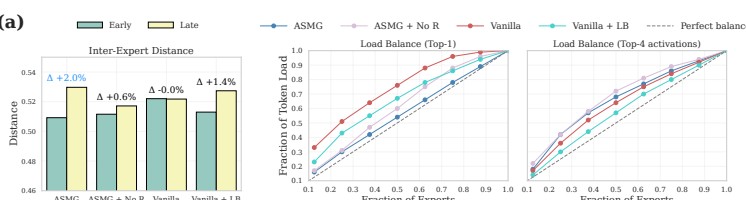

Figure 5: **Routing Specialization and Load Balance on GLUE.** (a) Normalized inter-expert embedding distances across Early and Late training phases. (b) Lorenz curves of expert–token assignment for Top-1/4 routing.

To assess load balance, we measure how evenly tokens are routed across experts by computing the fraction of Top-1 and Top-4 assignments each expert receives. We sort experts by how much

load they receive and plot the cumulative load as we add experts from the most- to least-utilized, forming the Lorenz curve. Curves closer to the diagonal indicate more uniform routing. As shown in Figure 5 (b), ASMG achieves the most balanced routing in the Top-1 setting, with its curve lying closest to the diagonal. For Top-4 activations, ASMG becomes more skewed than Vanilla + LB, yet still maintains a reasonable level of expert utilization despite having no explicit balance regularization. This demonstrates that ASMG naturally avoids severe load imbalance soley driven by its structure-aware routing unlike Vanilla method that heavily relies on load balance regularization.

## 4.4 ABLATION STUDIES

Table 4: Ablation results on components of ASMG under the $(8, 4)$ configuration on GLUE.

| Algorithms (K,k) | CoLA | MRPC | QNLI | MNLI | RTE | Average |
|---|---|---|---|---|---|---|
| **ASMG (8,4)** | 66.62±0.65 | 89.68±0.20 | 92.61±0.10 | 86.74±0.11 | 75.57±0.68 | 82.24 |
| **ASMG w/o $R$ (8,4)** | 64.57±0.82 | 90.12±0.50 | 92.54±0.04 | 86.38±0.06 | 72.56±1.77 | 81.23 |
| **ASMG w/o Interpolation (8,4)** | 66.02±1.32 | 89.56±0.31 | 92.39±0.17 | 86.51±0.12 | 73.53±2.45 | 81.60 |

To understand the practical role of the mixing matrix $R$, we ablate its contribution by removing the mixing step and directly use the principal GHA basis as gating vectors. Further, we also ablate the interpolation with learnable standard gating matrix and soley depend on GHA-driven unsupervised basis for routing.

As shown in Table 4, removing the mixing matrix $R$ consistently degrades performance across GLUE tasks, confirming that simply using the raw GHA basis limits expert expressiveness. This aligns with Figure 5, where the No-$R$ variant shows weaker expert separation and less balanced load distribution pattern. Since mixing matrix $R$ relaxes the fixed ordering of the GHA principal components, forming the routing basis that are not tied to the variance ranking of the raw basis, it is essential for constructing more flexible and evenly distributed routing directions within the GHA subspace. Likewise, relying solely on the GHA basis without interpolating a learnable gating head further reduces accuracy, reflecting the importance of task-dependent signals provided by the gradient-guided gating. Especially, the role of learnable gating head is most principal during the early phase of training where the GHA basis is still evolving. Together, these ablations show that both components are necessary for ASMG to achieve its full performance.

## 4.5 TEST TIME ADAPTATION

So far, all experiments have been conducted with GHA updates disabled at test time, leaving the gating basis fixed after training. In this section, we investigate the scenario where the unsupervised GHA updates are enabled during inference, allowing the router to adapt its basis to the test distribution in an online and unsupervised manner. Such adaptation is particularly important in out-of-distribution (OOD) scenarios, where test data deviates from training and a fixed gating basis may fail to generalize on the new structure. To demonstrate the effectiveness of ASMG in Test Time Adaptation (TTA), we conduct evaluations in two complementary OOD settings.

**GLUE-X for Language OOD Generalization** We further evaluate ASMG on GLUE-X (Yang et al., 2023), which extends the GLUE benchmark with corresponding OOD test sets for each task. Models are finetuned on the in-distribution (ID) task and then evaluated on its OOD variant. All results are averaged over three random seeds. As shown in Figure 6, ASMG achieves higher test accuracy than the MoE baseline with cosine router, same architecture with MoE in Table 3, on most OOD datasets (5 out of 7) with average improvement 0.65%p. Even with small

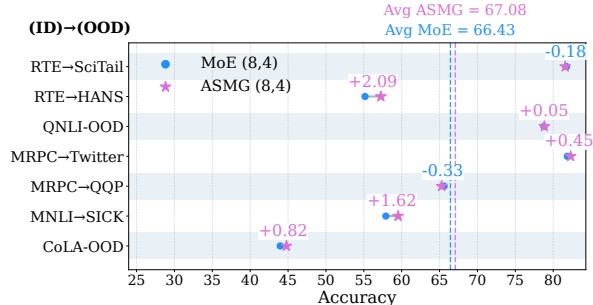

Figure 6: **OOD Performance comparison on GLUE-X.** The plot shows per-task accuracy differences between MoE and ASMG, along with average improvements.

margins, ASMG reliably improves over the MoE baseline, indicating that enabling test-time adaptation through structure-aware gating provides more robust handling of distribution shifts.

**ImageNet-C for Vision OOD**   We evaluate the test-time adaptation (TTA) capability of ASMG on ImageNet-C, a benchmark that applies 15 common image corruption types at multiple severity levels to ImageNet images (Hendrycks & Dietterich, 2019). We employ a ViT-S/32 (Dosovitskiy et al., 2021) model pretrained on ImageNet-1k (Dosovitskiy et al., 2021; Touvron et al., 2021) and apply MoEfication following the integration strategy of GMoE (Li et al., 2023), replacing the feed-forward blocks with MoE layers. Each MoE-augmented model is then finetuned on 20k randomly sampled images from ImageNet-1k under its corresponding routing method. After finetuning, evaluation is performed on ImageNet-C as an out-of-distribution (OOD) test set. For ASMG (TTA), we enable unsupervised GHA updates during inference so that the gating basis can adapt online to the corrupted test inputs in an unsupervised manner.

Table 5: Comparison of ImageNet-C top-1 accuracy (%) on ViT-S/32 across 15 corruption types at severity levels 1, 3, and 5.

| | Noise | | | Blur | | | | Weather | | | | Digital | | | | Avg |
|---|---|---|---|---|---|---|---|---|---|---|---|---|---|---|---|---|
| Model | Gauss | Shot | Impulse | Defocus | Glass | Motion | Zoom | Snow | Frost | Fog | Bright | Contrast | Elastic | Pixelate | JPEG | |
| SEVERITY 1 | | | | | | | | | | | | | | | | |
| **Vanila+LB** | 59.66 | 57.44 | 55.50 | 50.81 | 53.35 | 59.92 | 43.10 | 51.84 | 56.74 | 65.34 | 69.34 | 70.24 | 62.32 | 61.85 | 59.81 | 58.48 |
| **ASMG** | 60.67 | 58.84 | 57.16 | 51.08 | 53.74 | 60.72 | 44.85 | 51.91 | 56.60 | 64.86 | 69.57 | 69.86 | 62.92 | 62.08 | 59.57 | 58.96 |
| **ASMG (TTA)** | 60.24 | 58.56 | 56.24 | 51.23 | 54.20 | 60.87 | 45.55 | 51.94 | 57.21 | 65.24 | 69.80 | 69.98 | 62.78 | 62.13 | 60.08 | 59.07 |
| SEVERITY 3 | | | | | | | | | | | | | | | | |
| **Vanila+LB** | 42.94 | 38.74 | 39.77 | 32.67 | 21.36 | 39.69 | 27.97 | 32.69 | 30.46 | 54.92 | 65.95 | 65.72 | 55.31 | 49.21 | 54.56 | 43.46 |
| **ASMG** | 43.61 | 38.87 | 40.63 | 32.44 | 21.85 | 41.76 | 30.17 | 32.93 | 31.00 | 55.31 | 66.56 | 65.51 | 55.61 | 49.81 | 54.79 | 44.06 |
| **ASMG (TTA)** | 43.76 | 38.87 | 40.65 | 32.46 | 21.86 | 41.74 | 30.19 | 32.97 | 31.43 | 55.54 | 66.55 | 65.50 | 55.93 | 49.83 | 54.77 | 44.14 |
| SEVERITY 5 | | | | | | | | | | | | | | | | |
| **Vanila+LB** | 14.85 | 13.57 | 12.54 | 15.47 | 10.17 | 12.54 | 17.27 | 14.49 | 23.66 | 35.20 | 56.11 | 34.08 | 20.17 | 19.57 | 39.82 | 22.63 |
| **ASMG** | 14.80 | 12.91 | 12.46 | 14.98 | 10.91 | 19.22 | 18.59 | 16.85 | 23.74 | 36.52 | 57.95 | 36.54 | 21.05 | 21.36 | 40.28 | 23.88 |
| **ASMG (TTA)** | 14.80 | 13.33 | 12.49 | 16.09 | 10.94 | 19.22 | 18.55 | 16.94 | 24.72 | 37.78 | 57.96 | 36.52 | 21.10 | 21.80 | 40.57 | 24.19 |

The results in 5 shows that across all three corruption severities, ASMG improves over the Vanilla+LB baseline, and enabling test-time adaptation yields further gains. At modest corruption levels (severity 1 and 3), ASMG consistently improves over Vanilla+LB, and ASMG (TTA) achieves the highest average accuracies at both severities. Under the most challenging corruption level (severity 5), ASMG yields the largest improvement over Vanilla+LB, and ASMG (TTA) achieves the highest overall accuracy (24.19%). While TTA consistently provides additional gains across all severities, the improvements from ASMG alone already indicate that its structure-aware routing contributes meaningful OOD resilience even without test-time updates.

## 5 DISCUSSION

ASMG dynamically balances standard learnable gating and unsupervised structure-driven routing, leading to more reliable expert specialization without requiring auxiliary balancing losses. Across synthetic and real-data experiments, it consistently improves routing quality, maintains stable load usage, and enhances downstream performance. The method also supports optional test-time adaptation, enabling GHA updates during inference further reinforces the robustness under distribution shift, complementing the inherent OOD resilience already provided by ASMG.

One possible concern lies in the iterative nature of GHA updates, which may seem to increase the computational cost. However, our analyses in Appendix C analytically and empirically demonstrates that the extra overhead induced by ASMG remains negligible in practical MoE scenario. From time complexity to runtime measurements and peak memory usage, ASMG remains lightweight and compares favorably to existing MoE baselines. Overall, ASMG offers an efficient and specialization-focused routing mechanism that improves MoE stability and generalization across diverse settings.

## ETHICS STATEMENT

This work does not involve human subjects, personal data, or sensitive attributes. All experiments are conducted on publicly available datasets (GLUE, GLUE-X, DomainBed) and synthetic data that we generate procedurally, which are widely used in the research community. Our contribution is an algorithmic improvement to sparsely-gated Mixture-of-Experts models; we do not foresee ethical risks beyond those already present in standard neural network training and evaluation. We report compute and memory usage and favor modest training settings (single A6000 GPU) to encourage resource-aware experimentation. We will comply with licenses of all third-party datasets and code used, and we will release our implementation under a permissive license to support transparent, responsible research.

## REPRODUCIBILITY STATEMENT

We take reproducibility seriously and document all implementation details. Algorithmic specifics for ASMG (including the GHA updates and interpolation) are provided in Algorithm 1 (Appendix B) together with full complexity analysis (Appendix C). Exact training configurations and hyperparameters for every task are enumerated in Appendix F, including batch sizes, learning rates, MoE layer indices, iteration number $m$, and hardware. We fix and report random seeds and average all results over three seeds; data preprocessing, splits, and selection criteria (including the DomainBed train–validation selection rule) are described in Appendix F.1. Upon publication, we will release: (i) complete code for synthetic, GLUE/GLUE-X, and DomainBed experiments; (ii) configuration files and scripts to reproduce every table and figure. These materials will enable end-to-end replication of our results.

## THE USAGE OF LLMs.

In the preparation of this paper, we used large language models (LLMs) in a limited and supporting capacity. Specifically:

- **Writing aid and polishing:** LLMs were employed to improve the clarity, readability, and grammar of the manuscript. Their role was restricted to stylistic suggestions and refinement of phrasing, without altering the scientific content, claims, or conclusions.

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

## A  CODE AVAILABILITY

The source code for ASMG is available at `https://anonymous.4open.science/r/ASMG-6403`

## B  FULL ALGORITHMIC DETAILS OF ASMG

In this section, we provide the complete algorithms of the proposed Adaptive Structure-Aware MoE Gating (ASMG) method. The full forward-pass routine, including the iterative GHA updates and top-k expert aggregation, is summarized in Algorithm 1. ASMG begins by maintaining an orthonormal routing basis $\mathbf{V} \in \mathbb{R}^{K \times d}$, which is incrementally updated during training using the GHA. For each input $\mathbf{x} \in \mathbb{R}^d$, GHA performs $m$ iterations to refine $\mathbf{V}$ based on recent input directions. This evolving basis is then linearly mixed via a learnable coefficient matrix $\mathbf{R}$ to form a routing matrix $\mathbf{Z} = \mathbf{RV}$. Two sets of gating logits are then computed: one from the standard learnable gating matrix $\mathbf{W}_g$, and one from the structure-aware basis $\mathbf{Z}$. These two sources are combined through a learnable interpolation with elementwise gating weights $\alpha \in \mathbb{R}^K$ to form the final routing scores. Finally, the top-$k$ experts are selected based on these scores, and their outputs are aggregated accordingly.

---

**Algorithm 1** GHA-driven Adaptive Structure-Aware MoE Gating (**ASMG**)

---

1: **Given:** input $\mathbf{X} \in \mathbb{R}^{N \times d}$, trainable gating matrix $\mathbf{W}_g \in \mathbb{R}^{K \times d}$, GHA basis $\mathbf{V} \in \mathbb{R}^{K \times d}$, basis mixing matrix $\mathbf{R} \in \mathbb{R}^{K \times K}$, interpolation coefficients $\alpha \in \mathbb{R}^K$, expert matrices $\{\mathbf{W}_k\}_{k=1}^K$, iteration number per GHA update, $m$.

2: Randomly initialize $\mathbf{W}_g, \mathbf{V}, \mathbf{R}, \alpha, \{\mathbf{W}_k\}_{k=1}^K$

3: **for** each input $\mathbf{x} \in \mathbb{R}^d$ **do**

4:     GHA update with $m$ iterations per each forward pass:

5:     **for** $m$ iterations **do**

6:       **for** each $i$ in $\{1, ..., K\}$ **do**

7:         $y_i = \mathbf{v}_i^\top \mathbf{x}$

8:         $\mathbf{v}_i \leftarrow \mathbf{v}_i + \eta y_i \left( \mathbf{x} - \sum_{j=1}^i y_j \mathbf{v}_j \right)$

9:         $\mathbf{v}_i \leftarrow \mathbf{v}_i / \|\mathbf{v}_i\|_2$

10:       **end for**

11:     **end for**

12:     Basis mixing: $\mathbf{Z} = \mathbf{RV}$

13:     Compute routing probabilities:

$$l_{\text{vanilla}} = \mathbf{x}\mathbf{W}_g^\top, \quad l_{\text{GHA}} = \mathbf{x}\mathbf{Z}^\top$$
$$s = \text{Softmax}(\sigma(\alpha) \odot l_{\text{vanilla}} + (1 - \sigma(\alpha)) \odot l_{\text{GHA}}), \quad \sigma : \text{Sigmoid}, \quad s \in \mathbb{R}^K$$

16:     Top-$k$ expert aggregation:

$$\hat{\mathbf{y}} = \sum_{k \in \text{Top-}k(\{s_j\}_{j=1}^K)} s_k \cdot f_k(\mathbf{x}), \quad f_k(\mathbf{x}) = \mathbf{W}_k\mathbf{x}$$

18: **end for**

---

## C  COMPUTATION ANALYSIS

In this section, we analyze the computational complexity of ASMG in comparison to standard MoE gating mechanisms. We break down the cost of each stage, GHA basis updates, subspace mixing, and routing. The goal is to clarify the additional overhead introduced by GHA updates and demonstrate that it remains moderate relative to the overall cost of MoE forward and backward passes. Consider a mini-batch $X \in \mathbb{R}^{B \times d}$, hidden size $d$, number of experts (basis components) $K$, routing fan-out $k$ with top-$k$ selection, expert's width $d'$, and $m$ GHA iterations per pass.

**Standard Linear Gate.** Batch gate logits: $XW_g^\top$ costs $\mathcal{O}(BKd)$. The $k$ selected expert MLPs dominate with $\mathcal{O}(Bkdd')$. Thus the baseline batch cost is

**ASMG.** $\qquad\qquad\qquad\qquad \mathcal{O}(BKd) \ + \ \mathcal{O}(Bkdd').$

- GHA updates: Using the cached projection $Y = X\mathbf{V}^\top$ and prefix cumsums, each iteration updates all $K$ rows in $\mathcal{O}(BKd)$; with $m$ iterations: $\mathcal{O}(mBKd)$.

- Basis mixing: We form $\mathbf{Z} = \mathbf{RV}$ once per pass, which results in $\mathcal{O}(K^2 d)$.
- Extra logits: Given $\mathbf{Z}$, computing the additional logits $l_{\text{GHA}} = X\mathbf{Z}^\top$ costs $\mathcal{O}(BKd)$, and interpolation/softmax adds $\mathcal{O}(BK)$.

All other operations (softmax/top-$k$, expert MLPs) are unchanged from the baseline.

Therefore, the batch cost of ASMG is

$$\underbrace{\mathcal{O}\big(B((m{+}2)Kd)\big)}_{\text{GHA updates + extra logits}} + \underbrace{\mathcal{O}(K^2 d)}_{\text{mixing, once per pass}} + \underbrace{\mathcal{O}(Bkdd')}_{\text{experts}}.$$

Equivalently, the incremental overhead over the baseline is

$$\mathcal{O}((m+1)BKd) + \mathcal{O}(K^2 d).$$

Under the typical ASMG setup, we use $m \in \{1, 3\}$, $K \ll \min(d, d')$, we have

$$\mathcal{O}((m+1)BKd) + \mathcal{O}(K^2 d) \ll \mathcal{O}(Bkdd').$$

Thus, computation complexity added by ASMG is negligible compared to the dominant operations in the overall MoE architecture.

**Analysis of Runtime and Memory.** Table 6 reports runtime and memory usage statistics across different MoE models. For training, we measure (i) micro-step time, the latency of a single forward/backward micro-batch, (ii) optimization-step time, the latency per parameter update including gradient accumulation, and (iii) peak GPU memory usage during training. For inference, we report (i) latency per example, averaged over multiple runs after warmup, and (ii) peak GPU memory usage during inference.

We observe that ASMG achieves nearly identical or even smaller training cost to the MoE with cosine router, despite performing three iterative GHA updates per forward pass. This is because the cosine router itself requires additional computation for regularizing the similarity matrix, while GHA updates scale linearly with $K$ and $d$, resulting in negligible overhead. Consequently, ASMG provides improved routing performance without incurring extra cost during training. At inference time, when test-time GHA updates are disabled, ASMG incurs only minimal latency and memory overhead by reusing the fixed gating basis derived during training, effectively matching the efficiency of the baseline. If test-time GHA updates are enabled, inference latency increases due to additional online updates, but this is optional and provides robustness under distribution shift. Overall, ASMG offers a favorable trade-off, introducing no extra burden in training while supporting flexible test-time adaptation.

Table 6: Runtime and memory comparison. Times are mean±std (ms). Peak memory is in GiB.

| Method | Inference | | Training (per step) | | |
|---|---|---|---|---|---|
| | Latency ↓ | Peak Mem ↓ | Micro-step ↓ | Opt-step ↓ | Peak Mem ↓ |
| DynMoE | 17.2±9.1 | 5.2 | 428.4±168.7 | 80.2±11.9 | 11.4 |
| Vanilla | 10.2±7.2 | 0 4.7 | 127.9±95.2 | 43.6±2.7 | 7.9 |
| Vanilla + LB | 11.0±14.5 | 4.8 | 130.7±90.5 | 43.7±2.3 | 7.9 |
| MoE (cosine router) | 12.0±16.0 | 4.8 | 134.7±100.0 | 43.9±2.7 | 8.0 |
| ASMG (m = 3) | 10.3±5.5 | 4.7 | 129.1±94.2 | 43.7±2.8 | 7.9 |
| ASMG (TTA, m = 1) | 10.4±6.6 | 4.8 | 127.1±92.4 | 43.6±2.8 | 8.0 |
| ASMG (TTA, m = 3) | 16.4±7.6 | 4.8 | - | - | - |

## D  BASIS QUALITY ANALYSIS WITH VARYING GHA ITERATIONS

In this section, we analyze how the number of GHA update iterations per forward pass ($m$) affects the quality of the constructed basis. Since GHA operates incrementally and approximates the principal components through iterative updates, the choice of $m$ governs the trade-off between computational cost and approximation fidelity. We compare the cumulative explained variance of GHA-derived basis to that of full-batch SVD across three representative settings: (i) hidden states from the trained

encoder of Transformer used in the second synthetic task, (ii) ResNet-18 features on CIFAR-100, and (iii) ResNet-18 features on TinyImageNet. In each case, we vary $m \in \{1, 3, 10\}$ and evaluate the variance captured by the top 100 components.

Figure 7 shows that as the number of GHA iterations increases, the cumulative explained variance curves increasingly align with those obtained from SVD. This confirms that higher $m$ enables GHA to better approximate the principal subspace. Overall, GHA provides a viable online alternative to SVD, with approximation quality controllable via $m$. Note that GHA fits a fixed number of components $K$, and the cumulative variance curve is normalized over these $K$ dimensions. In contrast, SVD operates over the full rank of the input data (i.e., $d$), and reports the ratio of total variance explained up to the top-$K$ components. As a result, GHA curves always end at 1.0 by construction, while the corresponding SVD curves may saturate earlier or lower depending on how much of the total variance is captured within the top-$K$ singular directions.

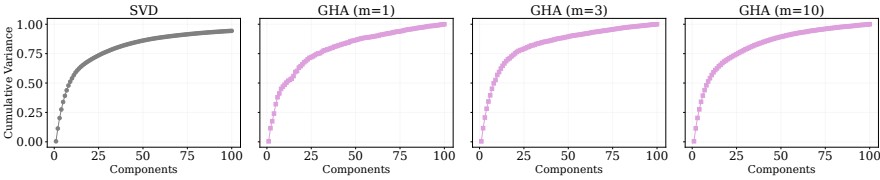

(i) Hidden states from the Transformer encoder used in the synthetic language task.

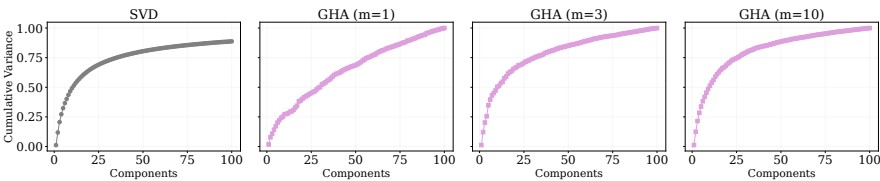

(ii) ResNet-18 features from CIFAR-100.

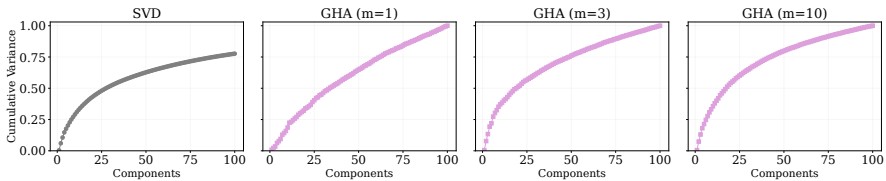

(iii) ResNet-18 features from TinyImageNet.

Figure 7: **Effect of GHA iteration number $m$ on basis approximation quality.** Cumulative explained variance of the top 100 components extracted using GHA with varying $m \in \{1, 3, 10\}$, compared to full-batch SVD. Results are shown for three settings: (i) Transformer hidden states on synthetic language datasets, (ii) ResNet-18 features on CIFAR-100, and (iii) ResNet-18 features on TinyImageNet. Higher $m$ improves approximation fidelity to SVD.

## E ABLATION STUDIES

We conduct ablation studies to analyze the sensitivity of ASMG to its key design choices. In particular, we focus on two key aspects of ASMG. First, we examine the effect of reducing the number of GHA updates $m$ performed at each training step. Since the iterative updates are designed to incrementally refine the principal subspace, varying $m$ allows us to understand how much adaptation is required to achieve stable and accurate routing. Second, we analyze the role of the GHA-driven basis itself in capturing the input structure and promoting expert specialization. To this end, we replace the learned principal basis with a fixed set of orthogonal random vectors while keeping the interpolation with the standard learnable gating matrix unchanged. This comparison disentangles the benefit of true data-driven subspace learning from simply mixing in an auxiliary random basis.

Table 7: Ablation study on the number of GHA iterations ($m$) in GLUE benchmark. Results are reported as Avg±Std over three random seeds.

| $m$ | $(K, k)$ | CoLA | MRPC | QNLI | MNLI | RTE | Average |
|---|---|---|---|---|---|---|---|
| 1 | (8, 1) | 64.68±0.83 | 90.52±0.15 | 92.40±0.18 | 86.47±0.03 | 74.49±0.45 | 81.71 |
| | (8, 2) | 66.49±0.47 | 89.85±1.12 | 92.58±0.06 | 86.66±0.13 | 74.85±1.96 | 82.10 |
| | (8, 4) | 65.36±1.17 | 90.10±0.35 | 92.40±0.15 | 86.48±0.14 | 75.33±1.19 | 81.94 |
| 2 | (8, 1) | 66.25±0.96 | 90.47±1.28 | 92.46±0.20 | 86.65±0.21 | 74.13±0.85 | 81.99 |
| | (8, 2) | 65.93±2.19 | 90.03±0.32 | 92.59±0.18 | 86.76±0.08 | 74.24±0.34 | 81.91 |
| | (8, 4) | 65.03±0.76 | 89.62±0.96 | 92.48±0.08 | 86.55±0.24 | 75.10±0.29 | 81.76 |
| 3 | (8,1) | 65.54±0.47 | 90.25±0.55 | 92.56±0.23 | 86.52±0.18 | 74.61±0.74 | 81.90 |
| | (8,2) | 65.81±0.80 | 90.03±0.32 | 92.52±0.15 | 86.63±0.08 | 75.57±0.61 | 82.11 |
| | (8,4) | 66.62±0.65 | 89.68±0.20 | 92.61±0.10 | 86.74±0.11 | 75.57±0.68 | 82.24 |

### E.1 PERFORMANCE WITH SMALLER GHA UPDATES

Table 7 reports results with smaller numbers of GHA iterations $m \in \{1, 2, 3\}$. While all settings consistently outperform the baseline MoE and DynMoE models, the choice of $m = 3$ yields the most consistent and superior performance across tasks. Smaller values of $m$ (e.g., $m = 1$ or $m = 2$) still provide competitive results, confirming that even limited iterative updates are sufficient to capture structural patterns, but additional updates stabilize training and improve generalization.

### E.2 PERFORMANCE WITH RANDOM BASIS

Table 8: Ablation study on random basis replacing the GHA-driven principal basis in GLUE benchmark. Results are reported as Avg±Std over three random seeds.

| Algorithm $(K, k)$ | CoLA | MRPC | QNLI | MNLI | RTE | Average |
|---|---|---|---|---|---|---|
| ASMG (8, 1) | 64.51±1.56 | 90.00±0.10 | 92.52±0.17 | 85.78±1.24 | 73.53±0.17 | 81.26 |
| ASMG (8, 2) | 65.18±0.28 | 89.10±1.30 | 92.46±0.04 | 69.48±24.06 | 74.73±2.06 | 78.19 |
| ASMG (8, 4) | 65.80±0.69 | 90.20±0.67 | 92.54±0.25 | 75.52±15.85 | 73.89±2.47 | 79.59 |

To further assess the importance of GHA-driven subspace learning, we replace the evolving principal subspace with fixed orthogonal random vectors, while keeping the interpolation with the learnable gating matrix unchanged. As shown in Table 8, replacing the GHA-driven principal basis with a fixed random orthogonal basis leads to a clear degradation in performance across all settings. Notably, the results reveal severe instability on MNLI, where the variance across seeds becomes extremely large (e.g., ±24.06 for $(8, 2)$), indicating that ASMG with random gating basis fails to converge to stable expert allocation. In contrast, the incremental subspace learning provided by GHA ensures both higher accuracy and stability, demonstrating that the gains of ASMG stem not simply from interpolating with an auxiliary gating basis, but from leveraging a data-driven, structure-aware representation that evolves with training.

## F EXPERIMENTAL SETUP

We present the experimental setup used to evaluate ASMG across language and vision tasks. This section first describes the datasets, covering the GLUE benchmark for language understanding and the DomainBed benchmark for domain generalization, and then outlines the training configurations and hyperparameters adopted for fair comparison with existing MoE baselines.

### F.1 DATASETS

**GLUE Benchmark (Language Tasks).** For language modeling, we follow the MoEfication setup Zhang et al. (2022) and finetune BERT-large Devlin et al. (2019) on the GLUE benchmark Wang et al. (2019). We evaluate on five representative GLUE subtasks: CoLA Warstadt et al. (2019) (linguistic acceptability), MRPC Dolan & Brockett (2005) (paraphrase detection), QNLI Wang et al. (2019) (question-answer entailment), MNLI Xu et al. (2020) (natural language inference), and RTE Bentivogli et al. (2009) (textual entailment). These datasets jointly cover grammaticality, semantic similarity, and entailment, providing a comprehensive testbed for expert specialization in language understanding.

**GLUE-X OOD Benchmark.** To evaluate robustness under distribution shift, we additionally consider GLUE-X Yang et al. (2023), an extension of GLUE that augments each in-distribution (ID) task with corresponding out-of-distribution (OOD) test sets drawn from different domains. GLUE-X provides 15 OOD datasets spanning eight GLUE tasks, enabling systematic assessment of cross-domain generalization. Importantly, models are trained only on the standard GLUE training sets and then directly evaluated on unseen OOD datasets, without exposure to target-domain data.

In our experiments, we follow the GLUE-X protocol and evaluate ASMG on selected OOD datasets corresponding to the five GLUE subtasks used in our ID experiments (CoLA, MRPC, QNLI, MNLI, and RTE). Specifically, we adopt CoLA $\rightarrow$ CoLA-OOD, MRPC $\rightarrow$ QQP / Twitter, QNLI $\rightarrow$ QNLI-OOD, MNLI $\rightarrow$ SICK, and RTE $\rightarrow$ SciTail / HANS, as shown in Figure 6.

**General Tasks for QWEN1.5-MoE Finetuning.** For finetuning the Qwen1.5-MoE backbone, we train on a mixed collection of downstream tasks spanning both enhancement and adaptation settings. For mathematical reasoning, we use GSM8K (Cobbe et al., 2021), consisting of grade-school word problems with short-form rationales. For code-related finetuning, we adopt CodeAlpaca and related evolutionary code instruction data (Chaudhary, 2023), evaluated on MBPP (Austin et al., 2021) and HumanEval (Chen et al., 2021). For intent understanding we use the BDCI-21 Smart HCI NLU dataset, which converts natural-language instructions into structured JSON outputs. For legal judgment prediction, we use the BDCI-21 Law Event Prediction dataset, providing civil case descriptions paired with judgment outcomes. For summarization, we adopt the BDCI-21 customer-service transcript summarization dataset. These datasets collectively cover instruction following, structured prediction, long-form summarization, legal reasoning, translation, and symbolic problem solving, forming a comprehensive evaluation suite for downstream specialization (Wang et al., 2024c).

**ImageNet-1k and ImageNet-C.** ImageNet-1k is a large-scale visual classification benchmark containing 1,000 object categories and 1.28M training images, serving as the pretraining source for our ViT-S/32 backbone. For robustness evaluation, we adopt ImageNet-C, which applies 15 corruption types spanning noise, blur, weather, and digital distortions, each at severity levels 1–5. These corruptions simulate realistic distribution shifts and enable systematic assessment of OOD robustness. Following the standard protocol, models are trained on clean ImageNet-1k and evaluated directly on ImageNet-C without exposure to corrupted images during training.

### F.2 HYPERPARAMETERS AND CONFIGURATION

For synthetic experiments, we provide the architectural choices, HMM generation parameters, MoE configurations, and GHA update rules used during training.

For language-domain evaluations, we cover four settings: (i) zero-shot evaluation on LLaMA-MoE (182M) MoE pretraining, (ii) LoRA-based finetuning on Qwen1.5-MoE, (iii) MoE-augmented BERT-large finetuning on GLUE, and (iv) OOD generalization on GLUE-X. For each configuration, we provide batch sizes, optimizer schedules, MoE layer placements, iteration number $m$, and all model-specific hyperparameters.

For vision-domain experiments, we detail the MoE integration into ViT-S backbones, the DomainBed training protocol, and the ImageNet-C evaluation procedure under both fixed-basis inference and test-time GHA updates.

Table 9: Pretraining configuration for the 182M LLaMA-MoE model. All hyperparameters exactly follow the ReMoE setup Wang et al. (2025).

| Category | Hyperparameter | Value |
|---|---|---|
| **Backbone (LLaMA-MoE 182M)** | | |
| | # layers | 12 |
| | Hidden size $d$ | 768 |
| | FFN hidden size | $4d = 3072$ |
| | Attention heads | 12 |
| | Query groups (GQA) | 4 |
| | Sequence length / max positions | 1024 / 1024 |
| | Dropout (attention / hidden) | 0.0 / 0.0 |
| **MoE CONFIGURATION** | | |
| | # experts $E$ | 8 |
| | Top-$k$ routing | $k = 1$ |
| | Router type | pre-softmax, ASMG |
| | $m$ GHA iteration | 1 |
| | GHA learning rate | $2 \times 10^{-5}$ |
| **OPTIMIZATION** | | |
| | GPU | $1 \times$ RTX 6000 Blackwell |
| | Global / micro batch size | 512 / 64 |
| | Base learning rate | $5 \times 10^{-4}$ |
| | Min learning rate | $5 \times 10^{-5}$ |
| | LR schedule | cosine decay |
| | LR warmup fraction | 0.01 |
| | Precision | bfloat16 |
| **DATA** | | |
| | Dataset | The Pile |
| | Tokenizer | GPT-2 BPE |
| | Training tokens | 30B (seq. 1024, global batch 512 for 60k steps) |

Table 10: Finetuning configuration for the 14B Qwen1.5-MoE backbone under the ESFT (Wang et al., 2024c) (LoRA-based) setup.

| Config | GSM8K | Code (MBPP/HE) | Intent | Law | Summary | Translation |
|---|---|---|---|---|---|---|
| **ASMG / ROUTING** | | | | | | |
| $m$ (GHA iterations) | 3 | 3 | 3 | 3 | 3 | 3 |
| Router update | full | full | full | full | full | full |
| LoRA on experts | yes | yes | yes | yes | yes | yes |
| **OPTIMIZATION** | | | | | | |
| Learning rate | $2 \times 10^{-4}$ | $2 \times 10^{-4}$ | $6 \times 10^{-4}$ | $5 \times 10^{-4}$ | $1 \times 10^{-6}$ | $2 \times 10^{-4}$ |
| LR schedule | cosine | cosine | cosine | cosine | cosine | cosine |
| Warmup ratio | 0.03 | 0.03 | 0.03 | 0.03 | 0.03 | 0.03 |
| Weight decay | 0.0 | 0.0 | 0.0 | 0.0 | 0.0 | 0.0 |
| Grad clip | 1.0 | 1.0 | 1.0 | 1.0 | 1.0 | 1.0 |
| Precision | bfloat16 | bfloat16 | bfloat16 | bfloat16 | bfloat16 | bfloat16 |
| **TRAINING** | | | | | | |
| Epochs | 3 | 3 | 3 | 3 | 3 | 3 |
| Train batch size / GPU | 4 | 4 | 4 | 4 | 4 | 4 |
| Eval batch size / GPU | 2 | 2 | 2 | 2 | 2 | 2 |
| Max length | 2048 | 2048 | 2048 | 2048 | 2048 | 2048 |
| Packing | enabled | enabled | enabled | enabled | enabled | enabled |
| **DATA** | | | | | | |
| Dataset | GSM8K | CodeAlpaca / MBPP / HE | Custom intent set | Law dataset | Summarization dataset | Translation dataset |
| Tokenizer | Qwen1.5 tokenizer | same | same | same | same | same |
| **HARDWARE** | | | | | | |
| GPU | | | $1 \times$ RTX 6000 Blackwell | | | |

Table 11: Detailed training hyper-parameters and configuration for GLUE fientuning experiments.

| Config | CoLA | MRPC | QNLI | MNLI | RTE |
|---|---|---|---|---|---|
| $m$ (iteration number) | 3 | 3 | 3 | 3 | 3 |
| Epoch | 10 | 7 | 3 | 3 | 9 |
| Learning rate | 2e-5 | {2e-5, 3e-5}$^*$ | 2e-5 | 2e-5 | {2e-5, 3e-5}$^*$ |
| MoE layer | {10, 12}$^*$ | 10 | {10, 12}$^*$ | 10 | {10, 12}$^*$ |
| LR schedule | Linear | Linear | Linear | Linear | Linear |
| Weight decay | 0.0 | 0.0 | 0.0 | 0.0 | 0.0 |
| Train Batch size / GPU | 32 | 32 | 32 | 32 | 32 |
| Eval Batch size / GPU | 8 | 8 | 8 | 8 | 8 |
| GPU | | | $1 \times$ A6000 (48G) | | |

Table 12: Training and evaluation configuration for the vison OOD experiments on ViT-S/32. We follow the GMoE integration strategy (Li et al., 2023).

| Category | Hyperparameter | Value |
|---|---|---|
| BACKBONE AND DATA | | |
| | Backbone | ViT-S/32 |
| | Pretraining dataset | ImageNet-1k |
| | OOD evaluation dataset | ImageNet-C (15 corruptions, severities 1, 3, 5) |
| MoE CONFIGURATION | | |
| | # experts $E$ | 6 |
| | Top-$k$ routing | $k = 2$ |
| | Iteration number $m$ (ASMG) | 3 |
| OPTIMIZATION | | |
| | GPU | $1 \times$ RTX A6000 |
| | Learning rate | $5 \times 10^{-5}$ |
| | Weight decay | 0.0 |
| | Dropout | 0.1 |
| | Batch size | 256 |
| | Training steps | 10,000 |

# G   FULL EXPERIMENTAL RESULTS ON SYNTHETIC-DATA

## G.1   FULL RESULTS ON THE EVOLUTION OF INTERPOLATION COEFFICIENT $\alpha$

We provide the full results on the dynamics of the interpolation coefficient $\alpha$ across all expert sizes $K$ and top-$k$ settings, extending the representative trends shown in the main paper. For each configuration, we compute (i) the slope of the mean $\alpha$ over epochs and (ii) the temporal evolution of $\alpha$ throughout training. Specifically, for each epoch we log per-expert $\alpha$ values, average them across experts and random seeds, and then fit a linear regression of $\alpha$ against epochs to obtain the slope. This slope reflects the overall direction of $\alpha$'s change during training. In addition, we plot the mean $\alpha$ at every epoch with standard deviation across seeds, showing the full trajectory of its evolution.

Figure 8 confirms the consistent downward trend of $\alpha$, observed across all $K$ and $k$ configurations. This indicates that the gating mechanism increasingly shifts toward reliance on the GHA-driven subspace as training progresses, reducing dependence on the purely learnable gating matrix. While the absolute changes in $\alpha$ are small in magnitude, the uniformity of the trend across settings demonstrates the robustness of ASMG's balance adaptation. These extended results reinforce our main finding that incremental subspace learning via GHA is progressively prioritized during training, leading to stable expert specialization.

# H   FULL EXPERIMENTAL RESULTS ON REAL-DATA

## H.1   LANGUAGE TASK ON GLUE BENCHMARK

Here, we report the full results on the GLUE benchmark for MoE models with expert size $K = 16$ across three top-$k$ configurations ($k = 1, 2, 4$). Table 13 compares the ASMG against MoE with cosine router across five GLUE datasets: CoLA (Warstadt et al., 2019), MRPC (Dolan & Brockett, 2005), QNLI (Wang et al., 2019), MNLI (Xu et al., 2020), and RTE (Bentivogli et al., 2009). Results show that GHA maintains competitive or superior average accuracy in all cases, with the largest

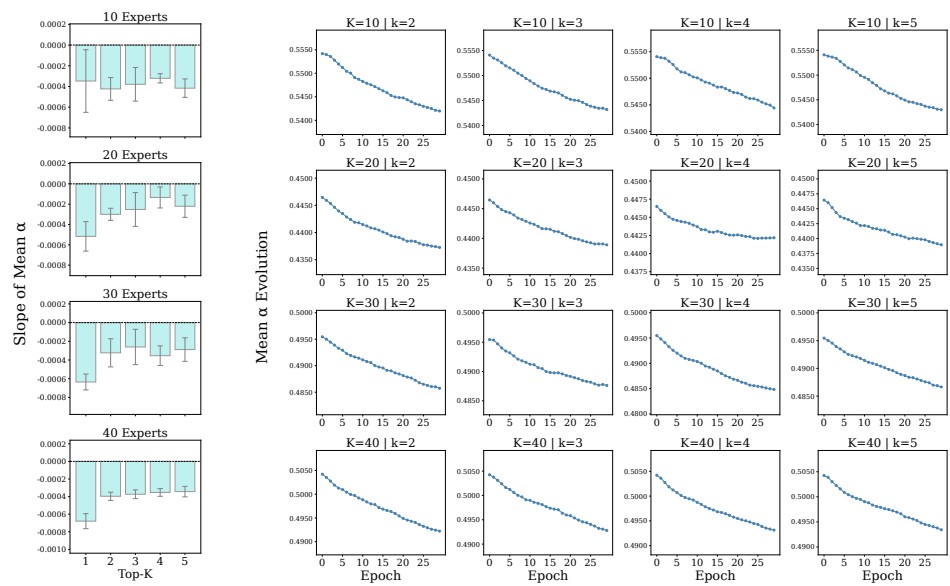

Figure 8: **Evolution of Interpolation Coefficient** $\alpha$**.** Left: Average slope of the mean $\alpha$ across different Top-$k$ settings. Right: Temporal evolution of mean $\alpha$ throughout training epochs.

gain observed at $(K = 16, k = 4)$. These results support the scalability and robustness of the proposed gating strategy in larger expert configurations. Table 11 shows the hyper-parameters and configurations used for language-task on GLUE dataset.

Table 13: **Performance of Vanilla MoE and DynMoE (% $\pm$ std) across five GLUE datasets and average accuracy. (K=16)**, [1] from Guo et al. (2025)

| Algorithms (K, k) | CoLA | MRPC | QNLI | MNLI | RTE | Average |
|---|---|---|---|---|---|---|
| MoE (16,1)[1] | 63.63±0.20 | 89.81±0.30 | 92.39±0.21 | 86.63±0.17 | 74.01±0.29 | 81.29 |
| MoE (16,2)[1] | 64.71±1.21 | 90.18±1.33 | 92.53±0.07 | 86.73±0.43 | 72.32±3.54 | 81.29 |
| MoE (16,4)[1] | 64.12±1.42 | 89.74±0.40 | 92.65±0.09 | 86.59±0.16 | 75.33±0.95 | 81.69 |
| **ASMG** (16,1) | 63.69±0.95 | 89.95±0.34 | 92.57±0.08 | 86.64±0.06 | 73.65±0.62 | 81.30 |
| **ASMG** (16,2) | 64.48±0.96 | 90.35±0.39 | 92.66±0.05 | 86.64±0.08 | 72.93±1.35 | 81.41 |
| **ASMG** (16,4) | 64.10±1.11 | 90.20±0.37 | 92.49±0.05 | 86.59±0.07 | 75.21±0.68 | 81.72 |

## H.2 TEST TIME ADAPTATION

We provide the complete results for ASMG with test-time adaptation (TTA), where unsupervised GHA updates are activated during both validation—for model selection—and test evaluation on unseen domains or tasks. This setup ensures that the adaptive gating basis remains responsive to the target distribution throughout the evaluation pipeline.

Table 14: Performance comparison on OOD test datasets from GLUE-X benchmark.

| Method $(K, k)$ | CoLA → CoLA-OOD | MRPC → QQP | MRPC → Twitter | QNLI → QNLI-OOD | MNLI → SICK | RTE → SciTail | RTE → HANS |
|---|---|---|---|---|---|---|---|
| MoE (8, 4) | 43.96 ± 3.79 | **65.60 ± 0.83** | 81.86 ± 0.17 | 78.78 ± 0.32 | 57.91 ± 1.07 | **81.76 ± 0.45** | 55.16 ± 0.44 |
| ASMG (8, 4) | **44.78 ± 1.30** | 65.27 ± 0.27 | **82.31 ± 0.23** | **78.83 ± 0.35** | **59.53 ± 1.07** | 81.58 ± 1.21 | **57.25 ± 2.43** |

For OOD experiments on GLUE-X, we reproduce the MoE baseline with cosine router under its best-performing configuration $(K, k) = (8, 4)$, and compare against ASMG with the same setting. The reproduced MoE achieves an average in-distribution accuracy of 81.37 on GLUE. When evaluated

on corresponding OOD test sets, Table 14 shows that ASMG outperforms the cosine router MoE baseline on 5 out of 7 datasets, with an average improvement of +0.65%p.

Table 15: DomainBed results with test-time GHA updates enabled.

| Method | PACS | VLCS | OfficeHome | TerraInc | Average |
|---|---|---|---|---|---|
| ASMG (Test-time GHA) | 88.5 | 80.6 | 73.9 | 49.4 | **73.1** |

For vision domain generalization experiment, results in Table 15 show that ASMG with test-time GHA achieves an average accuracy of 73.1 across four domain generalization datasets. Together, these results confirm that enabling test-time GHA updates allows ASMG to dynamically adjust its gating basis to the OOD distribution, improving both robustness under domain shifts.

