# OpenReview forum: "ASMG: Data Structure-Aware Routing via Incremental Subspace Learning for MoE"
_ICLR.cc/2026/Conference — Submitted to ICLR 2026_

### Official Review · Reviewer_9td1 · 2025-10-24

**Soundness:** 3
**Presentation:** 3
**Contribution:** 2
**Rating:** 4
**Confidence:** 4

**Summary:**

This paper explores a method for MoE gating by computing a gating weight matrix on the principal components of the inputs.  The (approximate) principal components are learned online using GHA, and the gating matrix decomposed into RV where R is in the principal component space, limited to K components.  This is combined with a vanilla gating matrix using linear combination with a learned combination parameter.  The behavior of this method is explored first on two synthetic toy data distributions (gaussian classification and HMM language stand-in), and then applied to five language tasks from GLUE and four vision datasets, with small but likely consistent improvements.

**Strengths:**

Limiting gating to principal components makes intuitive sense as a way to increase gating robustness, especially earlier on in training, as it could quickly align experts to work within buckets of the largest variance directions.  The approach has promise, with good analyses in the two synthetic toy sections.  The analyses in the toy data examples clearly illustrate a source of potential.

**Weaknesses:**

While the real-world data seems to have some consistent improvements, these are very small right now.  The mixture coefficient between plain gating and GHA gating goes towards GHA, this is also a small change (at least for the current training time and schedule).

It's also unclear how this interacts with balancing, and if the source improvements of this method is separate or overlapping from effects of balancing.  Balancing constraints for MoE are common and important, and these also lead to better expert specializations and usage, for example by reducing assignment collapse.  So it's important to understand how this method interacts with these techniques.  Does it have similar effects and reduce the need for them?  Or is it separate and a potential source of gain on top?

Overall, while there are some good ideas here, the work still seems preliminary.  Is the mixture really needed, or can it be reduced to the GHA side more aggressively (and can this or other small changes lead to larger gains)?  Does this method work with multiple expert layers (right now gains are small enough in the real-world experiments that it's unclear how much it's actually enabling itself)?

**Questions:**

3.2.3  says Z = RV is initialized with random R and "subsequently trained" (l.195), but then in the next sentence says "R is fixed and not updated".  This seems conflicting --- if R isn't updated, what is updated in Z?

3.2.5:  what is the cosine similarity computed over?  the text just says it's similarity between experts.  is it between gating vectors?  or something with the outputs, or expert weights?

3.3.4:  "Since hidden representations evolve dynamically throughout training, it is no longer feasible to directly optimize the full GHA-derived gating matrix Z".:  I don't see why this would necessarily be the case.  What are the issues that happen learning the GHA gating from the start?  also what is alpha initialized to?

Fig 6:  is the y axis alpha or sigmoid(alpha)?

* What are the distributions of assignments to experts for each strategy (naive, SVD/GHA, ASMG)?

---

> ### Author Response · Authors · 2025-11-22
>
> ## **[W1] Limited Experimental Results to Validate ASMG**
>
> - We sincerely appreciate the reviewer’s concern about the modest improvements observed in the presented real-world experiments and the need to more convincingly demonstrate the practical value of ASMG.
> - In response to the raised concerns, we substantially expanded the evaluation to include frontier MoE routing baselines and pretrained LLM-scale backbones, demonstrating that ASMG provides consistent and non-marginal gains even under strong recent routing methods.
> - Specifically, we
>     - (1) pretrain a LLaMA-MoE-182M model on 30B tokens from Pile and report zero-shot downstream performance
>     - (2) finetune the Qwen1.5-MoE-14B model under LoRA adaptation, comparing ASMG directly against Vanilla+LB under identical training conditions.
>     - These new results directly addresses the reviewer’s concern regarding the applicability and robustness of ASMG in large-scale and recent MoE architectures.
>
> ### **[W1.1.] Zero-shot Evaluation on Pretrained LLaMA-MoE (182M) (Table 1.)**
>
> To validate the effectiveness of our proposed method ASMG in LLM-style MoE architectures, we conduct the pretraining experiment following the experimental setup provided by NVIDIA Megatron-LM [1] and ReMoE [2]. We adopt a LLaMA-style decoder backbone and apply MoE structure with 182M active parameters out of 777M total parameters by replacing its feed-forward block with MoE layer with E=8 experts under Top-1 selection.
>
> We pretrain this LLaMA-MoE-182M model on 30B tokens from The Pile, and then perform zero-shot evaluation on commonsense reasoning and reading–comprehension benchmarks. As recommended, we compare ASMG against Expert-Choice (EC) routing [3], as well as recent strong MoE variants including Hash [4], Lory [5], dMoE [6], ReMoE [2], and SparseMixer-v2 [7].
>
> All models are trained from scratch under the same compute budget, and evaluated in a pure zero-shot setting across ARC-c, ARC-e, BoolQ, HellaSwag, LAMBADA, PIQA, and RACE. ($^1$ from [2])
>
> | Model           | ARC-c | ARC-e | BoolQ | HellaSwag | LAMBADA | PIQA | RACE | Avg.  |
> |-----------------|-------|-------|-------|-----------|---------|------|-------|-------|
> | Hash¹ [4]           | 19.28 | 45.45 | 54.95 | 29.68     | 31.44   | 63.06 | 27.66 | 38.79 |
> | Lory¹  [5]      | 20.31 | 42.97 | 49.54 | 28.75     | 32.35   | 62.24 | 27.75 | 37.70 |
> | SparseMixer-v2¹ [7] | 19.80 | 46.72 | 45.86 | 30.24     | 34.12   | 62.89 | 29.00 | 38.39 |
> | EC¹  [3]      | 18.86 | 42.97 | 60.21 | 29.14     | 29.26   | 61.92 | 27.37 | 38.53 |
> | dMoE¹ [6]    | 20.05 | 45.16 | 57.83 | 29.83     | 32.97   | 63.55 | 28.33 | 39.67 |
> | ReMoE¹ [2]    | 20.22 | 46.68 | 54.16 | 30.26     | 35.94   | 63.55 | 29.38 | 40.03 |
> | **ASMG**        | 20.06 | 48.86 | 59.66 | 30.43     | 33.27   | 64.69 | 28.52 | **40.78** |
>
> ASMG attains the **highest average zero-shot score, surpassing competitive routing approaches**. This improvement under a full LLaMA-style pretraining setup indicates that the gains from structure-aware gating persist beyond small synthetic settings and remain evident even in large-scale training regimes.
>
>
> ### **[W1.2.] Qwen1.5-MoE 14B Finetuning with LoRA Adaptation  (Table 2.)**
>
> To further validate ASMG in a full LLM-scale MoE setting, we next evaluate its performance during parameter-efficient finetuning on the 14B Qwen1.5-MoE backbone. Following the experimental protocol of ESFT [8], the router is fully updated for both Vanilla and ASMG, while all other modules, including expert, are finetuned through LoRA adaptation. We train on a diverse mixture of downstream tasks covering mathematical reasoning (GSM8K), code generation (CodeAlpaca), intent classification, legal judgment prediction, summarization, and translation. Evaluation is conducted on GSM8K, MBPP, HumanEval, as well as held-out splits for intent, law, summarization, and translation.
>
> Under this setup, we compare the standard Top-k sparse Vanilla MoE with load-balancing loss [9] (Vanilla + LB) against ASMG, ensuring identical training and optimization conditions.
>
> | Backbone            | Method        | GSM   | MBPP  | HE    | Intent | Law   | Summary | Translation | Avg   |
> |---------------------|---------------|-------|-------|-------|--------|--------|----------|-------------|--------|
> | Qwen1.5-MoE (14B)   | Base Model    | 36.77 | **38.40** | 33.54 | 16.83  | 13.90 | 21.40   | 14.26       | 25.01 |
> |                     | Vanilla + LB  | **42.60** | 33.61 | 34.75 | 65.60  | 25.70 | 30.70   | 29.36       | 33.09 |
> |                     | **ASMG**      | 42.16 | 35.48 | **39.02** | **68.40**  | **27.00** | **35.90**   | **29.60**      | **39.65** |
>
> ASMG achieves the highest overall average accuracy across all downstream tasks, improving upon the Vanilla+LB MoE by +6.56% on average. These results demonstrate that ASMG transfers effectively to full-scale LLM MoE backbones and provides consistent, non-marginal improvements under PEFT finetuning setup.

---

> ### Author Response · Authors · 2025-11-22
>
> ### **[W1.3.] OOD Robustness of ASMG on ImageNet-C (Table 5.)**
>
> - For rigorously controlled assessment of OOD robustness, we adopt ImageNet-C benchamrk, which provides
>     - 15 corruption types
>     - Multiple corruption severity levels (1, 3, 5)
>
> - This benchmark allows us to directly compare the OOD-robustness between Vanilla+LB and ASMG under identical training conditions while varying the difficulty of the OOD shift.
>
> ### **Experimental Setup**
> - We take a ViT-S/32 model pretrained on ImageNet-1k and then we apply the MoE conversion by replacing feed-forward block with MoE-MLP block following the MoEfication [3] framework.
> - Then we finetune the resulting MoE-ViT model on a 20k subset of ImageNet-1k. Evaluation is then performed on ImageNet-C.
>
> ### **Results**
>
> - Table 5: Comparison of ImageNet-C top-1 accuracy (%) on ViT-S/32 across 15 corruption types at severity levels 1, 3, and 5.
>
>     | **Severity** | **Model**     | **Gauss** | **Shot** | **Impulse** | **Defocus** | **Glass** | **Motion** | **Zoom** | **Snow** | **Frost** | **Fog** | **Bright** | **Contrast** | **Elastic** | **Pixelate** | **JPEG** | **Avg** |
>     |--------------|---------------|-----------|----------|-------------|-------------|-----------|------------|----------|----------|-----------|---------|------------|--------------|-------------|--------------|----------|---------|
>     | **1**        | Vanilla+LB    | 59.66 | 57.44 | 55.50 | 50.81 | 53.35 | 59.92 | 43.10 | 51.84 | 56.74 | 65.34 | 69.34 | 70.24 | 62.32 | 61.85 | 59.81 | 58.48 |
>     |              | ASMG          | 60.67 | 58.84 | 57.16 | 51.08 | 53.74 | 60.72 | 44.85 | 51.91 | 56.60 | 64.86 | 69.57 | 69.86 | 62.92 | 62.08 | 59.57 | **58.96** |
>     | **3**        | Vanilla+LB    | 42.94 | 38.74 | 39.77 | 32.67 | 21.36 | 39.69 | 27.97 | 32.69 | 30.46 | 54.92 | 65.95 | 65.72 | 55.31 | 49.21 | 54.56 | 43.46 |
>     |              | ASMG          | 43.61 | 38.87 | 40.63 | 32.44 | 21.85 | 41.76 | 30.17 | 32.93 | 31.00 | 55.31 | 66.56 | 65.51 | 55.61 | 49.81 | 54.79 | **44.06** |
>     | **5**        | Vanilla+LB    | 14.85 | 13.57 | 12.54 | 15.47 | 10.17 | 12.54 | 17.27 | 14.49 | 23.66 | 35.20 | 56.11 | 34.08 | 20.17 | 19.57 | 39.82 | 22.63 |
>     |              | ASMG          | 14.80 | 12.91 | 12.46 | 14.98 | 10.91 | 19.22 | 18.59 | 16.85 | 23.74 | 36.52 | 57.95 | 36.54 | 21.05 | 21.36 | 40.28 | **23.88** |
>
> - Across all corruption severities, ASMG consistently outperforms the Vanilla+LB baseline, demonstrating that its structure-aware routing is inherently more robust to distribution shift.
> - At lower severities (1 and 3), ASMG shows reliable improvements, and under the most challenging setting (severity 5), ASMG delivers the largest gain over Vanilla+LB.
> - Overall, these results highlight that ASMG provides stronger OOD resilience across most of the 15 corruption types and severity levels compared to baseline, indicating that the input structure-awareness can offer a reliable robustness boost under severe OOD conditions.

---

> ### Author Response · Authors · 2025-11-22
>
> ## **[W2] Load Balance Concern**
>
> - We firstly appreciate the reviewer’s insightful question about how ASMG interacts with load-balance mechanisms and whether its improvements stem from overlap with, or are independent of, traditional balancing effects.
> - Your concern is well-founded: in conventional MoE systems, load-balance regularizers (e.g., Switch Transformer) are essential for preventing load collapse, promoting expert specialization, and ensuring efficient expert usage.
>
> ### **[W2.1.] How ASMG algorithmically supports balanced expert usage**
>
> - ASMG’s routing mechanism is explicitly designed to distribute the representational capacity encoded in the GHA basis across all experts, rather than letting a few high-variance directions dominate the gating outputs.
> - Since the raw GHA basis is hierarchically ordered by variance (PC1 ≫ PC2 ≫ PC3 …), directly using it as gating vectors forces experts to inherit this uneven hierarchy.
> - The mixing matrix $R$ removes this dependence by rotating and re-combining the raw PC directions, creating a routing subspace that is not tied to the data’s variance ordering.
> - This design yields two benefits:
>     - Prevents dominance of certain experts paired with a few principal basis, avoiding the reviewer’s concern.
>     - Produces flexible, evenly distributed routing directions that preserve load balance while still being structure-aware.
>
> - Importantly, this balanced routing behavior emerges naturally from ASMG’s construction itself, ASMG does not use any explicit load-balance regularization.
>
> ### **[W2.2.] Empirical Evidence for Balanced Expert Utilization (Figure 4, 5)**
>
> To directly address this concern, we compare ASMG’s routing specialization and load-balance properties against the standard Vanilla + LB baseline (Top-k sparse selection with load-balancing loss). Our revised paper provides detailed load-balance analyses in Figure 4 (synthetic but controlled) and Figure 5 (real GLUE setting). These figures directly address the reviewer’s concern.
>
> - **Figure 4**: Expert Specialization and Load Balance Across Varying Experts Count $K$
>
>     - Specialization: ASMG maintains high mutual information between experts and the input target of high variance (property, $s$) as K increases (0.98 at K=40), while Vanilla+LB collapses its input specialization (0.24 at K=40).
>
>     - Load Balance: ASMG keeps stable normalized load entropy $H_{\text{norm}}$ across all $K$. In contrast, Vanilla+LB becomes increasingly imbalanced (falling to 0.17 at K=40).
>
>
> - **Figure 5**: Load Balance and Specialization on GLUE experiment setup
>
>     - Inter-expert token embedding distances (Fig. 5a):
>         - Under Top-2 selection setting, ASMG shows the largest Early → Late increase in embedding distances between tokens routed to different experts (+2.0%), indicating that the functional role of experts become more distinct over training without collapsing onto a few dominant experts.
>         - **Ablation on $R$**: ASMG + No R variant shows only a modest Early -> Late increase (+0.6%) in inter-expert token distance, indicating weaker specialization.
>
>     - Load Balance curves (Fig. 5b):
>         - Top-1 routing: ASMG’s Lorenz curve lies closest to the diagonal, demonstrating the most balanced token distribution.
>         - Top-4 routing: Although ASMG curve becomes slightly more skewed in the denser (Top-4) activation regime, it remains competitive with Vanilla+LB, which explicit uses load-balancing regularizer. This demonstrates that ASMG’s structure-aware routing continues to prevent severe load skew even under more complex multi-activation settings.
>         - **Ablation (No $R$)**: The balance curves become noticeably more skewed when $R$ is removed under both Top-1 and Top-4 settings, indicating that direct reliance on variance-ordered PCs leads to uneven expert utilization.
>
> - Together, our results consistently indicate that ASMG achieves stable and well-distributed expert usage without relying on any explicit load-balance loss, demonstrating that its behavior is distinct from, and not overlapped with, conventional balancing mechanisms.

---

> ### Author Response · Authors · 2025-11-22
>
> ## **[W3] Benefit of Hybrid (GHA + Learnable) Router (+ Multi-Layer Inclusion of ASMG)**
>
> We thank the reviewer for raising an important conceptual question about whether the interpolation between the GHA-driven routing basis and the standard learnable gating head is truly necessary, or whether ASMG could rely solely on the unsupervised GHA side.
>
> ### **Motivation of Interpolation**
>
> - The core motivation behind our interpolated design is that the two routing signals capture fundamentally different—and complementary—forms of information, and neither is sufficient on its own in realistic, evolving training dynamics.
> - **GHA-based routing**
>     - Offers unsupervised, structure-aware gating derived from the evolving input distribution.
>     - These components provide stable geometric cues, especially useful when the input repersentation stabilizes in the later phase of training, allowing experts to specialize along meaningful axes of variation.
> - **Learnable gating**
>     - In contrast, this captures task-driven supervision  aligned with the optimization objective, enabling the router to focus on features directly relevant for downstream prediction.
>
> - The interpolated design of ASMG combines these two strengths.
> - By learning coefficients that blend the naive logits and the GHA-derived logits, ASMG ensures that routing remains
>     - Task-responsive (via the learnable gate)
>     - Structure-aware (via GHA)
>     - Adaptively balanced between the two throughout training as the embedding space evolves. (you can check **Figure 8. in Appendix G**)
>
>
> ### **Ablation on ASMG**
>
> To determine whether ASMG truly depends on both the GHA-driven routing basis and the learnable gating head, we ablate each component and compare against the full ASMG. The ablation is conducted under 3 settings,
> - Vanilla MoE : removes GHA-based routing.
> - ASMG w/o R : removes basis mixing matrix $R$
> - ASMG w.o Interpolation : removes learnable gating head.
>
>
> | **Algorithms (K, k)**    | **CoLA**       | **MRPC**       | **QNLI**       | **MNLI**       | **RTE**         | **Average** |
> |-----------------------------------|----------------|----------------|----------------|----------------|-----------------|-------------|
> | ASMG (8,4)                    | 66.62 ± 0.65   | 89.68 ± 0.20   | 92.61 ± 0.10   | 86.74 ± 0.11   | 75.57 ± 0.68    | 82.24   |
> | Vanilla (8,4)   | 65.10 ± 1.10 | 89.20 ± 0.40 | 92.10 ± 0.08 | 86.10 ± 0.10 | 73.90 ± 1.50 | 81.48
> | ASMG w/o R (8,4)              | 64.57 ± 0.82   | 90.12 ± 0.50   | 92.54 ± 0.04   | 86.38 ± 0.06   | 72.56 ± 1.77    | 81.23  |
> | ASMG w/o Interpolation (8,4)  | 66.02 ± 1.32   | 89.56 ± 0.31   | 92.39 ± 0.17   | 86.51 ± 0.12   | 74.53 ± 2.45    | 81.80  |
>
>
> - Removing the mixing matrix R lowers accuracy the most (82.24 → 81.23), showing that raw GHA basis alone, without R, routing collapses toward high-variance PCs, weakening specialization and load balance.
>
> - Removing the interpolation mechanism also reduces performance (82.24 → 81.80), indicating that task-driven signals from the learnable gating head are essential—especially early in training before the GHA basis is well aligned.
>
> - Vanilla MoE performs similarly to the ablations but consistently below the full ASMG, confirming that neither the GHA pathway alone nor the learnable gating pathway alone is sufficient.
>
>
>
> ### **[W3.2.] ASMG Inclusion Across Multiple MoE Layers**
>
> - We appreciate the reviewer’s question regarding whether ASMG continues to be effective when applied across multiple MoE layers. We address this concern directly in the expanded experimental results.
>
> - In the newly added large-scale and vision-domain experiments (refer to the responses in **W1**):
>
>     - **W1.1, W1.2.** — LLaMA-MoE (182M) Pretraining (Table 1.),  Qwen1.5-MoE-14B finetuning (Table 2.):
>         - ASMG is applied to all MoE layers of a LLaMA-MoE (182M) and Qwen1.5-MoE-14B, and we observe consistent improvements over baselines, with highest average performance.
>
>     - **W1.3** — ImageNet-C OOD robustness (Table 5.):
>         - ASMG is inserted into multiple FFN blocks of a ViT-S/32 model following the MoEfication procedure
>             - 12 transformer layers in total, with 3 MoE-converted FFNs given as F (FFN) - $\cdots$ - S (MoE) – F – S – F – S – F.
>         - Across 15 corruption types and several severity levels, ASMG consistently improves performance, demonstrating stable gains even when deployed in multiple MoE layers.

---

> ### Author Response · Authors · 2025-11-22
>
> ### **[Q1] Clarification about “Z = RV is initialized with random R and subsequently trained” vs. “R is fixed and not updated”**
>
> Thank you for catching this ambiguity. Here is the precise clarification:
> - In the **first synthetic task only (Gaussian mixture classification)**—which has been fully removed from the revised paper—the router operated directly on raw inputs with no prior embedding layer.
> - Because the input distribution does not change at all during training in this simplistic toy setting, we extract the fully converged GHA basis $V$ as pre-processing step and it stays fixed throughout actual training.
> - For this single synthetic experiment, we used a fixed random $R$ for initialization and did not update it.
>
> However:
> - In all main experiments (Synthetic 2, GLUE, ImageNet-C, Qwen1.5-MoE-14B, etc.), the model is trained end-to-end, and the input representation evolves over time.
> - Therefore, ASMG uses a learnable $R$ across all these experiments
>
> Since the fisrt synthetic experiment has been removed from the revised paper, **$R$ is consistently learnable in all experiments presented**.
>
> ### **[Q2] What is the cosine similarity computed over in Section 3.2.5?**
>
> - Thank you for noting the ambiguity. To clarify, the cosine similarity in Section 3.2.5 is not computed on gating vectors or token-level outputs.
> - It is computed **between the learned FFN weight representations of the selected experts**.
>
> Concretely:
>
> - Each expert’s FFN is flattened into a single parameter vector.
>
> - For each input, we identify
>
>     - the Top-1 expert (highest routing score), and
>
>     - either the second most-activated expert (Co-Activated pair) or the lowest-probability expert (Top-Lowest pair).
>
>     - We then compute the cosine similarity between the FFN parameter vectors of these expert pairs.
>
> - This representation-level similarity measures how closely the experts’ learned functions align, enabling analysis of specialization and collaboration.
>
>
> ### **[Q3.1.] Why is it no longer feasible to directly optimize the full GHA-derived gating matrix $Z$, and what issues arise when learning GHA gating from the start?**
>
> - Appreciate the reviewer’s careful reading and we deeply understand why the statements in **Sections 3.2.3** (basis construction for synthetic task 1) and **3.3.4** (basis construction for synthetic task 2) from the pervious version of paper sounds conflicting.
> - The confusion stems entirely from the first synthetic task, where the routing input is the raw input vector with no embedding layer, making the input distribution static.
> - Only in the first synthetic toy experiment (now removed from the paper)
>     - The router input is the raw data vector, so we can compute the full principal component basis $V$ directly from the dataset before training.
>     - Then we fix $V$, $R$ to construct fixed $Z = RV$.
>     - Then we take $Z$ as just an "initialization" of the standard gating weight matrix (i.e., like Vanilla MoE).
>     - During training, we discarded the GHA mechanism entirely and optimized $Z$ using standard gradient descent, just like a conventional gating layer.
>
> ### **[Q3.2.] $\alpha$ Initialization**
>
> - $\alpha$ in initialized as zeros throughtout the entire experimental setups so that $\sigma(\alpha)$ starts from 0.5, equally balancing the contribution of the two gating modes.
>
>
> ### **[Q4] y-axis of Figure 6.**
>
> - Thank you for pointing out the ambiguity. The y-axis for the Figure 6. is driven from $\sigma(\alpha)$. We use $\sigma(\alpha)$ because it's the actual quantity that determines the interpolation balance between the vanilla vs. GHA routing logits during training.
>
> ### **[Q5] Distribution of assignments to experts**
>
> - We agree that understanding how different routing strategies distribute tokens across experts is important.
> - The complete load-balance analysis is provided in Figure 4 (synthetic HMM task) and Figure 5 (GLUE experiments).
> - This analysis is discussed in detail in our response to **[W2] Load Balance Concern**, where we explain how ASMG maintains balanced expert utilization even without explicit load-balancing loss.
> - Please refer to this answer first and let us know if you need any  additional quantitative summaries or visualizations.
>
>
> ---------------------------------
>
> **References**
>
> [1] Wang, Zihan, et al. "Let the expert stick to his last: Expert-specialized fine-tuning for sparse architectural large language models." arXiv preprint arXiv:2407.01906 (2024).
>
> [2] Shazeer, Noam, et al. "Outrageously large neural networks: The sparsely-gated mixture-of-experts layer." arXiv preprint arXiv:1701.06538 (2017).
>
> [3] Zhang, Zhengyan, et al. "Moefication: Transformer feed-forward layers are mixtures of experts." Findings of the Association for Computational Linguistics: ACL 2022. 2022.

---

> > ### Comment · Reviewer_9td1 · 2025-11-25
> >
> > Thanks for your responses.  The new Qwen and Llama experiments substantially strengthen the results for this method.  I also like the measurement comparing to routing without interpolation (just gha-projected) and the new balancing measures.  I've raised my score to 6.
> >
> > I had a couple relatively minor questions when I read the new version:
> >
> > * fig 5b ---  vanilla+LB has worse balancing than vanilla in these figures, which is inconsistent with the text saying it relies on LB.  do you have any thoughts about this?  also not impossible the curve colors are swapped; if so it would probably be good to double-check all the curves correspond to the right results including the asmg lines.
> >
> > * fig 5a --- I'm not sure why the "early" values are relevant here, as opposed to just the inter-expert distances by the end or over the course of training.  Especially since if the router has been initialized the same for all of them, and it hasn't learned much yet, then the distribs should all be the same at init for all methods.  (also, the early values for all the methods are mostly the same, except for vanilla, whose early value is a larger distance than the others)

---

> ### Author Response · Authors · 2025-11-26
>
> Thank you for the thoughtful follow-up and for increasing the score, we sincerely appreciate it. Your careful review has been extremely helpful in strengthening the paper.
>
> **Figure 5b**
> - you are absolutely right. After re-examining the plot, we found that the vanilla and vanilla+LB curves were accidentally reversed in the previous upload. Thank you for catching this. We have corrected the figure and re-uploaded the revised version. We also re-checked other routing-related plots, including the ASMG curves, to ensure their correctness.
>
> **Figure 5a**
> - Regarding Fig. 5a, the Early values can slightly differ across methods because they are not taken at strict initialization. Embeddings from each phase are collected during the first (Early) or last (Late) 100 training steps with 32 batch size. By that point, each model has already experienced a small amount of method-specific updates, including the additional load balanced regularization, so the token embeddings routed to each expert are no longer completely independent from method itself. This is why the embedding distances even during the early phase can differ across methods, although the routers begin from the same initialization.
>
> Again, we deeply appreciate your detailed feedbacks on our paper. Please let us know if you need any more clarification.

---

### Official Review · Reviewer_H4c4 · 2025-10-27

**Soundness:** 2
**Presentation:** 3
**Contribution:** 2
**Rating:** 4
**Confidence:** 4

**Summary:**

This paper introduces Adaptive Structure-Aware MoE Gating (ASMG), a new gating mechanism for MoE models designed to capture structural variations in the input, thereby addressing the limitations of conventional shallow linear routers. The method leverages the Generalized Hebbian Algorithm (GHA) to incrementally learn an evolving principal subspace of the input distribution during training and, optionally, at inference. The final routing mechanism is an interpolation between this data-driven subspace and a standard learnable gating matrix. Analyses on synthetic data, alongside experiments on real-world language and vision benchmarks, confirm that ASMG yields improvements over standard MoE baselines.

**Strengths:**

- The paper is well-written, clearly structured, and easy to follow.
- The application of GHA is novel, enabling the routing mechanism to adapt during both training and inference.
- A key strength of ASMG is its capacity for online adaptation at test time, which offers a practical solution for improving model robustness under distribution shifts.
- The empirical evaluation across both language and vision tasks  demonstrates that the proposed method surpasses conventional gating strategies.

**Weaknesses:**

- The motivation for introducing the mixing coefficient matrix $\mathbf{R}$ is not well-established. The paper would be strengthened by an ablation study demonstrating the necessity of this component, as well as a clearer intuitive explanation for why a learnable linear combination of principal components is preferable to using the components directly.


- The proposed method constructs its structure-aware basis using GHA to find the top-K principal components of the input distribution. This formulation inherently constrains the number of basis vectors (and thus experts, $K$) to be less than or equal to the input's dimensionality $d$, i.e. $K \leq d$. This constraint runs counter to a primary advantage of sparse MoE, which is the ability to scale the number of experts far beyond the model's hidden dimension. This limitation could hinder ASMG's applicability in scenarios that require a very large pool of experts.

- The first synthetic task, Gaussian mixture classification, may be overly simplistic. Given that the data is generated from linearly separable clusters and ASMG's router is initialized by extracting principal directions from the entire training set as a pre-processing step (Section 3.2.3), the problem is significantly simplified. Consequently, the superior performance of ASMG in this setting is expected and may not generalize. The validation would be more compelling if conducted on a synthetic task with non-linear decision boundaries or more complex data structures.

- Furthermore, the insightful analyses on expert routing, representation, and collaboration are confined to this simplistic synthetic task. The paper would be more impactful if these analyses were extended to the real-world language and vision experiments to demonstrate that similar specialization behaviors occur in more complex settings.

- The paper lacks a critical ablation study on the necessity of the interpolation mechanism itself. An analysis comparing the full ASMG model to variants using only the GHA-driven matrix or only the standard learnable gating matrix would be essential to quantify the benefits of the proposed hybrid approach.

**Questions:**

In addition to the points raised above, I have the following questions for the authors:

- While Section 3.2.3 mentions that $\mathbf{R}$ is a fixed random matrix for the first synthetic task, it is not explicitly stated whether $\mathbf{R}$ is fixed or learnable in all experiments. Could the authors clarify this? Additionally, have you investigated whether a learnable $\mathbf{R}$ would be more beneficial than a fixed one?

- How does the proposed mechanism ensure balanced expert utilization? This is a critical factor in MoE models to prevent expert under-utilization or collapse.

---

> ### Author Response · Authors · 2025-11-22
>
> We sincerely thank the reviewer H4c4 for the careful reading of our work and for the constructive and detailed feedback. Your comments touch on several important aspects of ASMG, including the motivation for the mixing matrix $R$, the simplicity of the synthetic experiment, the need for clearer ablations and algorithmic clarification, and concerns regarding balanced expert utilization.
>
> ## **[Q1] Algorithm Clarification on $R$**
>
> - The confusion arises because we set $R$ fixed **only in the first synthetic task**, where the router directly consumed raw input vectors with no prior embedding or feature tranformation.
> - In this very specific and unrealistic scenario, the input distribution is static, so the GHA basis $V$ and the mixing matrix $R$ naturally remain fixed. However, this setting has now been removed entirely from the revised paper.
> - In all remaining experiments, including the multinomial HMM synthetic setup, Qwen1.5-MoE finetuning, GLUE/GLUE-X, and ImageNet-C, the ASMG algorithm follows the unified formulation presented in propoosed method (Section 3.3.4). Therefore, both $V$ and $R$ are dynamically updated throughout training in reponse to evolving feature representation.
>
> ## **[W1, Q2] Motivation Behind Mixing Matrix $R$ + Load Balance of ASMG**
>
> ### **Limitation of Direct Alignment**
>
> - The key issue of directly aligning PC to gating vectors is that raw GHA principal components are strictly ordered by input variance, meaning, PC1 ≫ PC2 ≫ PC3 …, directly using it as gating vectors forces experts to inherit this uneven hierarchy.
> - The mixing matrix $R$ removes this dependence by rotating and re-combining the raw PC directions, creating a routing subspace that is not tied to the data’s variance ordering.
>
> ### **Theoretic View**
>
> - GHA is an online algorithm for incrementally solving PCA. As the input stream $x$ evolves, GHA updates a set of directions that converge to the principal components of the data covariance, the PCA components.
> - Formally, let $V$ = $[v_1, v_2, ..., v_K]$ denote the matrix of GHA-driven principal components, ordered by decreasing variance, $\lambda_1 \gg \lambda_2 \gg \cdots \gg \lambda_K$ where $\lambda_i = \mathbb{E}[(x^\top v_i)^2]$ is the eigenvalue associated with component $v_i$.
>     - Derivations for $\lambda_i = \mathbb{E}[(x^\top v_i)^2]$.
>         - In PCA, the eigenvalue-eigenvector pair is defined through the covariance matrix $\sum = \mathbb{E}[xx^\top]$ via the relation $\sum v_i = \lambda_i v_i$.
>         - For any zero-mean input vector $x$, the projection onto the $i$-th principal direction is $s_i = x^\top v_i$ whose variance is $\mathbb{E}[s_i^2]$ = $\mathbb{E}[(x^\top v_i)^2]$.
>         - This can be rewritten as $\mathbb{E}\!\left[(x^\top v_i)^2\right] = \mathbb{E}\!\left[v_i^\top x x^\top v_i\right] = v_i^\top\, \mathbb{E}[x x^\top]\, v_i = v_i^\top \Sigma\, v_i.$
>         - Substituting $\Sigma v_i = \lambda_i v_i$ yields $v_i^\top \Sigma v_i = v_i^\top (\lambda_i v_i) = \lambda_i (v_i^\top v_i)$.
>         - Since each principal component is normalized, $v_i^\top v_i = 1$.
>         - Thus, $\lambda_i = \mathbb{E}\!\left[(x^\top v_i)^2\right]$, which is the variance of the data projected onto $v_i$.
>
> - If we directly use the principal components as gating vectors, then for an input token embedding $x$, the routing score assigned to expert $i$ is $s_i(x) = \sigma(x^\top v_i)$, where $\sigma$ denotes an activation function such as softmax or sigmoid. Since these functions are strictly increasing, they preserve the ordering.
>     - $x^\top v_i > x^\top v_j \quad\Longleftrightarrow\quad  \sigma(x^\top v_i) > \sigma(x^\top v_j). \quad \forall\,\, i, j \in [1, K]$
>
> - Therefore, the Top-k experts under the activated scores $\{s_i(x)\}$ are exactly the same as the Top-k experts under the raw projections $\{x^\top v_i\}$.
>
> - Hence, to analyze which experts are selected by the router, it is sufficient to examine the unactivated scores $x^\top v_i$.
>
> - Taking expectations over the magnitude of the raw projections, we have
>     - $\mathbb{E}[|s_i(x)|] = \mathbb{E}[\,|\sigma(x^\top v_i)|  \propto \mathbb{E}[\,|x^\top v_i|\,] \approx \sqrt{\lambda_i},$
>
> - Given the analysis above, the magnitude of each projection $x^\top v_i$ scales with $\sqrt{\lambda_i}$, and the eigenvalues satisfy $\lambda_1 \gg \lambda_2 \gg \cdots \gg \lambda_K$.
> - Since $\sigma$ preserves ordering, the activated scores inherit this hierarchy $\mathbb{E}[|s_1(x)|] \gg \mathbb{E}[|s_2(x)|] \gg \cdots \gg \mathbb{E}[|s_K(x)|]$.
>
> - Consequently, under Top-$k$ sparse routing, experts associated with high-variance components (e.g., $v_1, v_2$) dominate the selection process.
> - In other words, the variance ordering of the principal components induces a systematic expert–dominance effect, making direct PC-to-expert alignment inherently imbalanced.

---

> ### Author Response · Authors · 2025-11-22
>
> ### **Empirical Results (Figure 5. and Table 4.) on GLUE experiment setup**
>
> - **Figure 5**: Load Balance and Specialization
>     - Figure 5 provides empirical evidence that the mixing matrix $R$ is essential for achieving both strong specialization and balanced expert utilization.
>     - Inter-expert token embedding distances (Fig. 5a):
>         - Under Top-2 selection setting, ASMG shows the largest Early → Late increase in embedding distances between tokens routed to different experts (+2.0%), indicating that the functional role of experts become more distinct over training without collapsing onto a few dominant experts.
>         - **Ablation on $R$**: ASMG + No R variant shows only a modest Early -> Late increase (+0.6%) in inter-expert token distance, indicating weaker specialization.
>
>     - Load Balance curves (Fig. 5b):
>         - Top-1 routing: ASMG’s Lorenz curve lies closest to the diagonal, demonstrating the most balanced token distribution.
>         - Top-4 routing: ASMG remains competitive with Vanilla+LB, even though ASMG does not use any explicit load-balance loss. This shows that structure-aware routing naturally avoids heavy load skew.
>         - **Ablation on $R$**: The Lorenz curves become noticeably more skewed when $R$ is removed, indicating that direct reliance on variance-ordered PCs leads to uneven expert utilization, the exact issue raised in the review.
>
>
>     - Combining the results, the No-$R$ ablation exhibits weaker specialization and significantly more skewed balance curves, demonstrating that direct use of variance-ordered PCs leads to uneven and less-differentiated expert utilization.
>     - Mixing matrix $R$ is essential for breaking the strict variance hierarchy and allowing ASMG to maintain stable specialization and load balance.
>
>
> - **Table 4 : Performance Comparison from Ablation on Basis Mixing Matrix R**
>
>     | Algorithms (K, k)              | CoLA          | MRPC          | QNLI          | MNLI          | RTE            | Average |
>     |-------------------------------|---------------|---------------|---------------|---------------|----------------|---------|
>     | **ASMG (8,4)**                | 66.62 ± 0.65  | 89.68 ± 0.20  | 92.61 ± 0.10  | 86.74 ± 0.11  | 75.57 ± 0.68   | 82.24 |
>     | **ASMG w/o R (8,4)**          | 64.57 ± 0.82  | 90.12 ± 0.50  | 92.54 ± 0.04  | 86.38 ± 0.06  | 72.56 ± 1.77   | 81.23 |
>
>
>     - The ablation further clarifies how the basis mixing is important to fully leverage the performance of ASMG.
>     - Removing the mixing matrix $R$ consistently reduces accuracy across all GLUE tasks (average drops from 82.24 → 81.23). This degradation reflects that without $R$, the router relies on the raw, variance-ordered GHA components, leading to weaker expert separation and load imbalance

---

> ### Author Response · Authors · 2025-11-22
>
> ## **[W2]$\,$ K $\ll$ d Constraint in Computational Analysis**
>
> We appreciate the reviewer’s insightful observation regarding the constraint $K \ll d$.
> - We investigated whether this bound limits practical applicability by surveying the expert configurations of widely used modern MoE systems—including DeepSeek-MoE [1], DeepSeek-V2 [2], Qwen1.5-MoE [3], LLaMA-MoE [4], Mixtral-8×22B [5].
> - Across these production-level large scale MoE LLMs, the number of experts per layer ranges from 8 to at most 160, while their hidden dimensions typically lie between 2k and 8k.
> - Consequently, the ratio ${K}/{d_{\text{model}}}$ in these models typically falls below 1–4%, operating in a regime where $K  \ll d_{\text{model}}$.
> - We provide a comprehensive case-study table that investigates the expert / dimension ratio for the widely adopted recent LLM-MoE frameworks.
>
>
> | Model                 | $d_{\text{model}}$ | Experts per layer ($K_{\text{total}}$) | Active experts | $K_{\text{total}} / d_{\text{model}}$ |
> | --------------------- | ------- | ------------------------------- | ------------------------- | ----------------- |
> | DeepSeekMoE-16.4B [1] | 2048    | **64** routed + 2 shared  | top-6 routed + all shared | **3.2%**          |
> | DeepSeek-V2 [2]   | 5120    | **160** routed + 2 shared | top-6 routed + all shared | **3.2%**          |
> | Qwen1.5-MoE-A2.7B [3] | 2048    | **60** experts                  | top-4                     | **2.9%**          |
> | LLaMA-MoE-v1-3.5B (2/8) [4] | 4096    | **8** experts                   | top-2                     | **0.2%**          |
> | Mixtral-8×22B [5]  | 6144    | **8** experts                   | top-2                     | **0.13%**         |
>
>
> We appreciate the reviewer for raising this practical point for us to examine, and we found that, across widely deployed MoE systems, the expert counts used in practice remain below 1–4% of the model dimension, well within the $K \ll d$ range.
>
>
> -----------------------------------
>
> ## **[W3, W4] Analysis of ASMG Routing Behavior in More Practical Setup (Figure 4. and 5.)**
>
> We thank the reviewer for highlighting the need to assess beyond the overly simple Gaussian-mixture toy example and to examine ASMG’s routing behavior under more realistic, nonlinear, and high-dimensional conditions. We fully agree that richer synthetic structure and real-world language data provide a more reliable testbed for evaluating the practical behaviors of ASMG.
>
> To directly address this, we expand our analysis to two more practically complex settings:
>
> 1. A more challenging synthetic setup modeled by multinomial HMMs that carry latent entity–property states, probabilistic transitions, and a mixture of multiple HMM dynamics, yielding nonlinear decision boundaries and mixed contextual structure while remaining analytically interpretable.
>
> 2. Full real-world GLUE language modeling, where embeddings are high-dimensional, non-linearly separable, and governed by complex semantic structure.
>
>
> ### **Figure 4**: Routing Specialization and Load Balance Across Expert Counts
>
> Figure 4 evaluates how specialization and expert utilization behave as the number of experts $K$ grows in the synthetic sequence-modeling scenario.
>
> - Specialization: ASMG maintains high mutual information $I(e;s)$ between experts and the input target of high variance (property, $s$) as K increases (0.98 at K=40), while Vanilla+LB collapses its input specialization (0.24 at K=40).
>
> - Load Balance: ASMG keeps stable normalized load entropy $H_{\text{norm}}$ across all $K$. In contrast, Vanilla+LB becomes increasingly imbalanced (falling to 0.17 at K=40).
>
> ### **Figure 5**: Routing Behavior on Real-World GLUE Tasks
>
> - As discussed earlier when motivating the role of the mixing matrix R in **[W1, Q2]**, the results in Figure 5 further confirm the high specialization and balanced load distribution patterns of ASMG.
>
> - Figure 5 (a) : The strengthened input–expert specialization seen in synthetic Gaussian classfication task appears again in GLUE as a clear Early -> Late increase in inter-expert token embedding distance, showing that ASMG continues to sharpen expert roles even with high-dimensional, nonlinearly structured language inputs.
>
> - Figure 5 (b) : Moreover, this improved specialization naturally translates into  balanced utilization of experts based on well-differentiated functional role of them. Compared to vanilla or load-balanced baselines, ASMG maintains more uniform token allocation without relying on explicit load-balance regularization.
>
> Overall, Figure 4 and 5 demonstrate that the behavioral advantages identified in the higly controlled synthetic setting persist in more complex scenarios, validating the effectiveness of ASMG’s structure-aware routing in practical environments.

---

> ### Author Response · Authors · 2025-11-22
>
> ## **[W5] Extensive Ablation on ASMG Components (Table 4.)**
>
> - We appreciate the reviewer for suggesting to conduct ablation study on the interpolation mechanism to isolate the contribution of the structure-aware gating from standard learnable gating.
> - We completely agree that evaluating ASMG against the ablated variants that remove either the GHA-driven basis or the learnable gating matrix is critical for rigorously quantifying the benefits of our hybrid approach.
> - Since the ablation of GHA basis-dependent routing reduces to Vanilla gating with no load balance regularization, we report the Vanilla performance under the identical experimental settings.
>
>
> | **Algorithms (K, k)**    | **CoLA**       | **MRPC**       | **QNLI**       | **MNLI**       | **RTE**         | **Average** |
> |-----------------------------------|----------------|----------------|----------------|----------------|-----------------|-------------|
> | ASMG (8,4)                    | 66.62 ± 0.65   | 89.68 ± 0.20   | 92.61 ± 0.10   | 86.74 ± 0.11   | 75.57 ± 0.68    | 82.24   |
> | Vanilla (8,4)   | 65.10 ± 1.10 | 89.20 ± 0.40 | 92.10 ± 0.08 | 86.10 ± 0.10 | 73.90 ± 1.50 | 81.48
> | ASMG w/o R (8,4)              | 64.57 ± 0.82   | 90.12 ± 0.50   | 92.54 ± 0.04   | 86.38 ± 0.06   | 72.56 ± 1.77    | 81.23  |
> | ASMG w/o Interpolation (8,4)  | 66.02 ± 1.32   | 89.56 ± 0.31   | 92.39 ± 0.17   | 86.51 ± 0.12   | 74.53 ± 2.45    | 81.80  |
>
> - The ablation results in Table 4 show that both the mixing matrix $R$ and the interpolation mechanism are fundamental to achieve ASMG's full performance.
> - Removing $R$ lowers the average score from 82.24 → 81.23, confirming that directly using variance-ordered PCs for gating vectors limits expert specialization capacity of ASMG.
> - Removing the interpolation mechanism reduces performance (82.24 → 81.80), indicating that GHA alone cannot capture the task-dependent signals provided by the learnable gating head.
> - Together, these findings demonstrate that ASMG’s gains arise from the complementary roles of $R$ and interpolation, both of which are necessary for stable specialization and balanced expert utilization of ASMG.
>
>
> ------------------------------------------
> **Reference**
>
> [1] Dai, Damai, et al. "Deepseekmoe: Towards ultimate expert specialization in mixture-of-experts language models." arXiv preprint arXiv:2401.06066 (2024).
>
> [2] Liu, Aixin, et al. "Deepseek-v2: A strong, economical, and efficient mixture-of-experts language model." arXiv preprint arXiv:2405.04434 (2024).
>
> [3] Ahmed, Imtiaz, et al. "Qwen 2.5: A comprehensive review of the leading resource-efficient llm with potentioal to surpass all competitors." Authorea Preprints (2025).
>
> [4] Zhu, Tong, et al. "Llama-moe: Building mixture-of-experts from llama with continual pre-training." arXiv preprint arXiv:2406.16554 (2024).
>
> [5]-1 Jiang, Albert Q., et al. "Mixtral of experts." arXiv preprint arXiv:2401.04088 (2024).
>
> [5]-2 “Mistral AI. “Cheaper, Better, Faster, Stronger.” Mistral AI, 17 Apr. 2024, https://mistral.ai/news/mixtral-8x22b.

---

> > ### Comment · Reviewer_H4c4 · 2025-11-25
> >
> > Thank you for your detailed response. The authors have largely addressed my main concerns in the rebuttal. Please incorporate the above discussion into the final revision. Accordingly, I have raised my score to 6.

---

> > > ### Author Response · Authors · 2025-11-26
> > >
> > > Thank you very much for the positive follow-up and for raising the score. We appreciate your careful reading and constructive feedback throughout the process. We will make sure to incorporate the corresponding clarifications and discussions into the final revision.

---

### Official Review · Reviewer_tzUN · 2025-11-01

**Soundness:** 3
**Presentation:** 3
**Contribution:** 3
**Rating:** 6
**Confidence:** 4

**Summary:**

Focusing on specialized MoE routing, this paper proposes a novel interpolation method via iterative GHA. It shows good performance in various tasks and great OOD adaptation. The research problem is significant in MoE community, but there are no experiments on LLMs, which weakens the contributions.

**Strengths:**

- The research question of specialized gating is critical to MoE models.
- Although the iterative update costs additional computations, it brings test-time adaptation for OOD settings.
- The overall performance is good and brings new ideas to the MoE field.

**Weaknesses:**

- It’s a pity that this paper only conduct experiments on encoder-only models (e.g. BERT). It would be much more significant if the authors could conduct experiments on small-sized MoE models for language modeling.
    - Or, if it is impossible for you to train a model from scratch, could you convert a well-trained MoE model to GHA gating?
    - For example, OLMoE-1B-7B is a good start, and it would be better to extend to DeepSeek-V2-Lite, and Qwen3-30B-A3B (2~10B tokens would converge if only routers are trained). I understand the GHA would be compatible with current LLMs.
    - We (or at least me) really don’t care about finetuning-style GLUE at all in such an LLM era.
- The computational analysis may be biased. Although the most computational cost lies in the expert forward pass, GHA ( O(B((m+2)Kd)) + O(K^2d) ) is greater than the vanilla routing ( O(BKd) ). And your baseline should be vanilla MoE instead of DynMoE and cosine gate in Table 3.
- Algorithm 1 should be placed in the main content. The whole bunch of texts in section 3.1 really do not help readers understanding the whole process.

**Questions:**

- line 53: develope → develop
- line 101~104, \eta in the equation is not properly defined.
- What if the GHA is not utilized during inference? (i.e. set \sigma(\alpha) to 1.0 in evaluating GLUE benchmarks)

---

> ### Author Response · Authors · 2025-11-22
>
> ## **[W1] Limited Experimental Scope**
>
> - We sincerely thank the reviewer tzUN for pointing out the importance of evaluating ASMG in autoregressive LLM-style MoE settings. We fully agree that demonstrating the scalability of our routing mechanism beyond encoder-only models such as BERT would strengthen the contribution of the work. In the revised submission, we substantially expand the evaluation to include frontier MoE routing baselines and pretrained LLM-scale backbones, demonstrating that ASMG provides consistent and non-marginal gains even under strong recent routing methods.
>
> - Specifically, we
>     - (1) pretrain a LLaMA-MoE-182M model on 30B tokens from Pile and report zero-shot downstream performance
>     - (2) finetune the Qwen1.5-MoE-14B model under LoRA adaptation, comparing ASMG directly against Vanilla+LB under identical training conditions.
>
> ### **[W1.1.] Zero-shot Evaluation on Pretrained LLaMA-MoE (182M) (Table 1.)**
>
> To validate the effectiveness of our proposed method ASMG in LLM-style MoE architectures, we conduct the pretraining experiment following the experimental setup provided by NVIDIA Megatron-LM [1] and ReMoE [2]. We adopt a LLaMA-style decoder backbone and apply MoE structure with 182M active parameters out of 777M total parameters by replacing its feed-forward block with MoE layer with E=8 experts under Top-1 selection.
>
> We pretrain this LLaMA-MoE-182M model on 30B tokens from The Pile, and then perform zero-shot evaluation on commonsense reasoning and reading–comprehension benchmarks. As recommended, we compare ASMG against Expert-Choice (EC) routing [3], as well as recent strong MoE variants including Hash [4], Lory [5], dMoE [6], ReMoE [2], and SparseMixer-v2 [7].
>
> All models are trained from scratch under the same compute budget, and evaluated in a pure zero-shot setting across ARC-c, ARC-e, BoolQ, HellaSwag, LAMBADA, PIQA, and RACE. ($^1$ from [2])
>
> | Model           | ARC-c | ARC-e | BoolQ | HellaSwag | LAMBADA | PIQA | RACE | Avg.  |
> |-----------------|-------|-------|-------|-----------|---------|------|-------|-------|
> | Hash¹ [4]           | 19.28 | 45.45 | 54.95 | 29.68     | 31.44   | 63.06 | 27.66 | 38.79 |
> | Lory¹  [5]      | 20.31 | 42.97 | 49.54 | 28.75     | 32.35   | 62.24 | 27.75 | 37.70 |
> | SparseMixer-v2¹ [7] | 19.80 | 46.72 | 45.86 | 30.24     | 34.12   | 62.89 | 29.00 | 38.39 |
> | EC¹  [3]      | 18.86 | 42.97 | 60.21 | 29.14     | 29.26   | 61.92 | 27.37 | 38.53 |
> | dMoE¹ [6]    | 20.05 | 45.16 | 57.83 | 29.83     | 32.97   | 63.55 | 28.33 | 39.67 |
> | ReMoE¹ [2]    | 20.22 | 46.68 | 54.16 | 30.26     | 35.94   | 63.55 | 29.38 | 40.03 |
> | **ASMG**        | 20.06 | 48.86 | 59.66 | 30.43     | 33.27   | 64.69 | 28.52 | **40.78** |
>
> ASMG attains the **highest average zero-shot score, surpassing competitive routing approaches** including EC, Lory, and SparseMixer-v2. This improvement under a full LLaMA-style pretraining setup indicates that the gains from structure-aware gating persist beyond small synthetic settings and remain evident even in large-scale training regimes.
>
> ### **[W1.2.] Qwen1.5-MoE 14B Finetuning with LoRA Adaptation (Table 2.)**
>
> To further validate ASMG in a full LLM-scale MoE setting, we next evaluate its performance during parameter-efficient finetuning on the 14B Qwen1.5-MoE backbone. Following the experimental protocol of ESFT [8], the router is fully updated for both Vanilla and ASMG, while all other modules, including expert, are finetuned through LoRA adaptation. We train on a diverse mixture of downstream tasks covering mathematical reasoning (GSM8K), code generation (CodeAlpaca), intent classification, legal judgment prediction, summarization, and translation. Evaluation is conducted on GSM8K, MBPP, HumanEval, as well as held-out splits for intent, law, summarization, and translation.
>
> Under this setup, we compare the standard Top-k sparse Vanilla MoE with load-balancing loss [9] (Vanilla + LB) against ASMG, ensuring identical training and optimization conditions except the router choice.
>
> | Backbone            | Method        | GSM   | MBPP  | HE    | Intent | Law   | Summary | Translation | Avg   |
> |---------------------|---------------|-------|-------|-------|--------|--------|----------|-------------|--------|
> | Qwen1.5-MoE (14B)   | Base Model    | 36.77 | **38.40** | 33.54 | 16.83  | 13.90 | 21.40   | 14.26       | 25.01 |
> |                     | Vanilla + LB  | **42.60** | 33.61 | 34.75 | 65.60  | 25.70 | 30.70   | 29.36       | 33.09 |
> |                     | **ASMG**      | 42.16 | 35.48 | **39.02** | **68.40**  | **27.00** | **35.90**   | **29.60**      | **39.65** |
>
> ASMG achieves the highest overall average accuracy across all downstream tasks, improving upon the Vanilla+LB MoE by +6.56% on average. These results demonstrate that ASMG transfers effectively to full-scale LLM MoE backbones and provides consistent, non-marginal improvements under PEFT finetuning setup.

---

> ### Author Response · Authors · 2025-11-22
>
> ## **[W2] Computational Analysis (Table 6. in Appendix)**
>
> - We appreciate the reviewer for pointing out to include Vanilla MoE in computation comparison analysis. In response, we have expanded our runtime and memory study to include Vanilla, Vanilla+LB, DynMoE, and the cosine router, providing a more complete picture of ASMG’s computational footprint (Table 6).
> - All measurements are conducted under the same GLUE fine-tuning setup using the (8,4) MoE configuration (except the DynMoE with adaptive routing), where each method shares the identical expert architecture, batch size, optimizer, and micro-batch schedule.
>
> ### **Runtime and memory comparison (mean ± std). Peak memory in GiB.**
>
> - **Inference**
>     | Method              | Latency ↓   | Peak Mem ↓ |
>     | ------------------- | ----------- | ---------- |
>     | DynMoE              | 17.2 ± 9.1  | 5.2        |
>     | Vanilla             | 10.2 ± 7.2  | 0.47       |
>     | Vanilla + LB        | 11.0 ± 14.5 | 4.8        |
>     | MoE (cosine router) | 12.0 ± 16.0 | 4.8        |
>     | ASMG (m = 3)        | 10.3 ± 5.5  | 4.7        |
>     | ASMG (TTA, m = 1)   | 10.4 ± 6.6  | 4.8        |
>     | ASMG (TTA, m = 3)   | 15.4 ± 7.6  | 4.8        |
>
>
> - **Training (per step)**
>
>     | Method              | Micro-step ↓  | Opt-step ↓  | Peak Mem ↓ |
>     | ------------------- | ------------- | ----------- | ---------- |
>     | DynMoE              | 428.4 ± 168.7 | 80.2 ± 11.9 | 11.4       |
>     | Vanilla             | 127.9 ± 95.2  | 43.6 ± 2.7  | 7.9        |
>     | Vanilla + LB        | 130.7 ± 90.5  | 43.7 ± 2.3  | 7.9        |
>     | MoE (cosine router) | 134.7 ± 100.0 | 43.9 ± 2.7  | 8.0        |
>     | ASMG (m = 3)        | 129.1 ± 94.2  | 43.7 ± 2.8  | 7.9        |
>     | ASMG (TTA, m = 1)   | 127.1 ± 92.4  | 43.6 ± 2.8  | 8.0        |
>     | ASMG (TTA, m = 3)   | –             | –           | –          |
>
> - Under this controlled environment, the extended results show that ASMG’s training cost is nearly identical to Vanilla+LB and the cosine router, and that its inference latency matches standard MoE routing when test-time GHA is disabled.
> - This addresses the reviewer’s concern by demonstrating that ASMG during training process does not introduce hidden computational overhead relative to widely used Vanilla structure.
>
> ---------------
>
> ## **[W3] Algorithm Inclusion in main paper**
>
> - Thank you for highlighting the importance of clearer algorithm presentation. We agree that placing clear algorithmic overivew inside main paper would improve readability. We'll make sure to include the concise algorithmic overview directly in the main paper as the final revision version.
>
> ------------------------
>
> ## **[Q1-Q3] Typos and Algorithm Clarification**
>
> **[Q1]** : Thank you for catching this. We will correct “develope” → “develop” in the final revision.
>
> **[Q2]** :
> - You are absolutely right that $\eta$ was not properly defined in the text. In all experiments, $\eta$ denotes the learning rate that controls the step size of the incremental GHA updates. We have now added this definition to the experimental setup section.
> - We fix $\eta$=2×e−5 across entire experimental setup.
> - As shown in Figure 7 (Appendix D), this update rate yields a stable approximation of the evolving PCA subspace, validated through the tight alignment between the online GHA basis and the full SVD basis.
>
> **[Q3]** : At inference time, we simply disable GHA updates and use the fixed gating basis learned during training. This is completely equivalent to setting the interpolation weight $\sigma(\alpha) = 1$ and evaluating ASMG in its train-time–only configuration.

---

> > ### Comment · Reviewer_tzUN · 2025-11-27
> >
> > Thanks for the response! It's interesting to find that ASMG consumes similar training step time against Vanilla + LB. Although pre-training for 30B tokens is still in a performance fluctuation stage, I think ASMG has shown its potentials. I would like to maintain my positive rating.
> >
> > If you could conduct full-param SFT (instruction tuning) on OLMoE with ASMG and Vanilla+LB and demonstrate its effectiveness, I would like to further raise my scores, and I think this would change r2nH's mind too (if r2nH is not absent).

---

> > > ### Author Response · Authors · 2025-11-27
> > >
> > > Thank you very much for the constructive follow-up and for your positive assessment of the work. We truly appreciate your suggestion regarding full-parameter SFT on OLMoE with ASMG and Vanilla+LB. We agree that this evaluation would further strengthen the empirical validation.
> > >
> > > We are currently running the recommended experiment, following your suggested setup, to compare ASMG and Vanilla+LB under full-parameter instruction tuning on OLMoE. Once the runs are completed and the results stabilize, we will promptly share the findings here.
> > >
> > > Thank you again for the constructive guidance, we will keep you updated as soon as the experiments conclude.

---

> > > > ### Comment · Reviewer_tzUN · 2025-11-27
> > > >
> > > > Thanks for the quick feedback~ Looking forward to your new results!

---

> > > > > ### Author Response · Authors · 2025-11-29
> > > > >
> > > > > Following your suggestion, we conducted full-parameter instruction tuning (SFT) on the OLMoE (7B) backbone, comparing ASMG against Vanilla+LB. The results are summarized in the table below, where ASMG achieves an +0.89%p improvement in average performance over Vanilla + LB. These results show that ASMG continues to provide measurable benefits under full-param instruction-tuning settings.
> > > > >
> > > > > | Backbone             | Method                 | GSM   | MBPP  | HE    | Intent | Law   | Summary | Translation | Avg   |
> > > > > |---------------------|------------------------|-------|-------|-------|--------|-------|---------|-------------|-------|
> > > > > | **OLMoE (7B)**      | Base                   | 15.09 | 21.80 | 10.98 | 0.40   | 7.10  | 8.40    | 18.47       | 12.62 |
> > > > > |                     | Vanilla + LB (Full)    | 45.34 | 24.20 | 14.02 | 74.20  | 23.01 | 34.01   | 31.76       | 35.22 |
> > > > > |                     | **ASMG (Full)**        | **46.47** | **23.60** | **15.85** | **74.00** | **25.00** | **33.40** | **34.42** | **36.11** |

---

> ### Author Response · Authors · 2025-11-23
>
> **References**
>
> [1] Shoeybi, Mohammad, et al. "Megatron-lm: Training multi-billion parameter language models using model parallelism." arXiv preprint arXiv:1909.08053 (2019).
>
> [2] Wang, Ziteng, Jun Zhu, and Jianfei Chen. "Remoe: Fully differentiable mixture-of-experts with relu routing." arXiv preprint arXiv:2412.14711 (2024).
>
> [3] Zhou, Yanqi, et al. "Mixture-of-experts with expert choice routing." Advances in Neural Information Processing Systems 35 (2022): 7103-7114.
>
> [4] Roller, Stephen, Sainbayar Sukhbaatar, and Jason Weston. "Hash layers for large sparse models." advances in neural information processing systems 34 (2021): 17555-17566.
>
> [5] Zhong, Zexuan, et al. "Lory: Fully differentiable mixture-of-experts for autoregressive language model pre-training." arXiv preprint arXiv:2405.03133 (2024).
>
> [6] Gale, Trevor, et al. "Megablocks: Efficient sparse training with mixture-of-experts." Proceedings of Machine Learning and Systems 5 (2023): 288-304.
>
> [7] Liu, Liyuan, et al. "Grin: Gradient-informed moe." arXiv preprint arXiv:2409.12136 (2024).
>
> [8] Wang, Zihan, et al. "Let the expert stick to his last: Expert-specialized fine-tuning for sparse architectural large language models." arXiv preprint arXiv:2407.01906 (2024).
>
> [9] Shazeer, Noam, et al. "Outrageously large neural networks: The sparsely-gated mixture-of-experts layer." arXiv preprint arXiv:1701.06538 (2017).

---

### Official Review · Reviewer_ieFN · 2025-11-01

**Soundness:** 3
**Presentation:** 3
**Contribution:** 3
**Rating:** 4
**Confidence:** 4

**Summary:**

The paper proposes a novel gating mechanism for mixture-of-experts (MoE) layer which interpolates standard and principal component-aware gating mechanisms.  Generalized Hebbian Algorithm is used to approximate top-K PCs of data and allows online updates during the inference time as well. Through synthetic and real data benchmarks, they show that the proposed method can be more reliable compared to standard baselines.

**Strengths:**

1. **Clarity:** the method and motivation behind the method is quite clearly presented.
2. **Originality:** data structure-aware routing using GHA is an interesting touch to routing in MoE which also enables test-time adaptation at an acceptable computational cost.
3. **Significance:** Although the OOD performance improvement does not seem very substantial, I believe it is a right step forward in that robustness direction in terms of methodology.

**Weaknesses:**

1. **Motivation behind $R$:** The authors introduce a learnable mixing parameter $R \in \mathbb{R}^{K \times K}$. I think it is not sufficiently well-motivated AND/OR the explanation is a bit misleading.
- (i) Why don't we want direct alignment scores with PCs and rely on the naive gating term in terms of task specific alignment?
- (ii) Line 156 *"This creates a latent gating basis that spans the same routing subspace"*... this claim may not be true in general since $span(RV) \subseteq span(V)$ where equality holds iff $R$ is full-rank (which doesn't seem to be enforced). Would enforcing full-rankness of $R$ improve performance then?
2. **Analyzing the effect of test-time GHA:** As mentioned in Strengths section above, I think the test-time adaptation of $V$ is an attractive approach to OOD. However, I believe the OOD performance may not always prefer adapting $V$ to the test input over using a fixed $V$ from the pretraining stage. There are two ways to make this claim more convincing:
- (i) by including both train-time and test-time GHA versions in Figure 8. This is because the improvement in Figure 7 is not quite conclusive on its own since the improvement is modest and only average performance is reported.
- (ii) by testing on robustness benchmarks which has a *corruption strength* parameter such as ImageNet-C [1] and showing that test-time GHA performance degrades at a slower rate as the corruption severity increases.

___

Overall, I believe the paper proposes an interesting design to structure-aware and adaptive MoE gating mechanism with promising empirical results. Therefore, I am open to increase my score upon satisfactory responses to the concerns raised in the Weaknesses section.

___

### References

1. Dan Hendrycks, Thomas Dietterich. Benchmarking Neural Network Robustness to Common Corruptions and Perturbations. ICLR 2019

**Questions:**

See Weaknesses.

---

> ### Author Response · Authors · 2025-11-22
>
> We deeply appreciate the reviewer ieFN for the careful reading of our work and the constructive feedback. Your comments highlight two central concerns: (1) the motivation and role of the mixing matrix $R$ and (2) the need for clearer analysis of test-time GHA adaptation and validate the OOD robustness of ASMG on ImageNet-C with multi-level corruption severity.
>
> ## **[W1] Concerns for Basis Mixing Matrix $R$**
>
> ### **[W1.1] Why not direct alignment of PCs to gating vectors?**
>
> #### **Limitation of Direct Alignment**
>
> - The key issue of directly aligning PC to gating vectors is that raw GHA principal components are strictly ordered by input variance, i.e., PC1 ≫ PC2 ≫ PC3 …
> - If experts are aligned to these variance-ordered components, the expert tied to the dominant direction (PC1) would receive disproportionately many tokens, while experts aligned with lower-variance PCs would receive very few.
> - This inevitably results in expert dominance and load imbalance, making direct PC alignment unsuitable for MoE routing.
>
> #### **Theoretic View**
>
> - GHA is an online algorithm for incrementally solving PCA. As the input stream $x$ evolves, GHA updates a set of directions that converge to the principal components of the data covariance, the PCA components.
> - Formally, let $V$ = $[v_1, v_2, ..., v_K]$ denote the matrix of GHA-driven principal components, ordered by decreasing variance, $\lambda_1 \gg \lambda_2 \gg \cdots \gg \lambda_K$ where $\lambda_i = \mathbb{E}[(x^\top v_i)^2]$ is the eigenvalue associated with component $v_i$.
>     - Derivations for $\lambda_i = \mathbb{E}[(x^\top v_i)^2]$.
>         - In PCA, the eigenvalue-eigenvector pair is defined through the covariance matrix $\sum = \mathbb{E}[xx^\top]$ via the relation $\sum v_i = \lambda_i v_i$.
>         - For any zero-mean input vector $x$, the projection onto the $i$-th principal direction is $s_i = x^\top v_i$ whose variance is $\mathbb{E}[s_i^2]$ = $\mathbb{E}[(x^\top v_i)^2]$.
>         - This can be rewritten as $\mathbb{E}\!\left[(x^\top v_i)^2\right] = \mathbb{E}\!\left[v_i^\top x x^\top v_i\right] = v_i^\top\, \mathbb{E}[x x^\top]\, v_i = v_i^\top \Sigma\, v_i.$
>         - Substituting $\Sigma v_i = \lambda_i v_i$ yields $v_i^\top \Sigma v_i = v_i^\top (\lambda_i v_i) = \lambda_i (v_i^\top v_i)$.
>         - Since each principal component is normalized, $v_i^\top v_i = 1$.
>         - Thus, $\lambda_i = \mathbb{E}\!\left[(x^\top v_i)^2\right]$, which is the variance of the data projected onto $v_i$.
>
> - If we directly use the principal components as gating vectors, then for an input token embedding $x$, the routing score assigned to expert $i$ is $s_i(x) = \sigma(x^\top v_i)$, where $\sigma$ denotes an activation function such as softmax or sigmoid. Since these functions are strictly increasing, they preserve the ordering.
>     - $x^\top v_i > x^\top v_j \quad\Longleftrightarrow\quad  \sigma(x^\top v_i) > \sigma(x^\top v_j). \quad \forall\,\, i, j \in [1, K]$
>
> - Therefore, the Top-k experts under the activated scores $\{s_i(x)\}$ are exactly the same as the Top-k experts under the raw projections $\{x^\top v_i\}$.
>
> - Hence, to analyze which experts are selected by the router, it is sufficient to examine the unactivated scores $x^\top v_i$.
>
> - Taking expectations over the magnitude of the raw projections, we have
>     - $\mathbb{E}[|s_i(x)|] = \mathbb{E}[\,|\sigma(x^\top v_i)|  \propto \mathbb{E}[\,|x^\top v_i|\,] \approx \sqrt{\lambda_i},$
>
> - Given the analysis above, the magnitude of each projection $x^\top v_i$ scales with $\sqrt{\lambda_i}$, and the eigenvalues satisfy $\lambda_1 \gg \lambda_2 \gg \cdots \gg \lambda_K$.
> - Since $\sigma$ preserves ordering, the activated scores inherit this hierarchy $\mathbb{E}[|s_1(x)|] \gg \mathbb{E}[|s_2(x)|] \gg \cdots \gg \mathbb{E}[|s_K(x)|]$.
>
> - Consequently, under Top-$k$ sparse routing, experts associated with high-variance components (e.g., $v_1, v_2$) dominate the selection process.
> - In other words, the variance ordering of the principal components induces a systematic expert–dominance effect, making direct PC-to-expert alignment inherently imbalanced.

---

> ### Author Response · Authors · 2025-11-22
>
> #### **Empirical Results (Figure 5. and Table 4.) on GLUE experiment setup**
>
> - **Figure 5**: Load Balance and Specialization
>     - Figure 5 provides empirical evidence that the mixing matrix $R$ is essential for achieving both strong specialization and balanced expert utilization.
>     - Inter-expert token embedding distances (Fig. 5a):
>         - Under Top-2 selection setting, ASMG shows the largest Early → Late increase in embedding distances between tokens routed to different experts (+2.0%), indicating that the functional role of experts become more distinct over training without collapsing onto a few dominant experts.
>         - **Ablation on $R$**: ASMG + No R variant shows only a modest Early -> Late increase (+0.6%) in inter-expert token distance, indicating weaker specialization.
>
>     - Load Balance curves (Fig. 5b):
>         - Top-1 routing: ASMG’s Lorenz curve lies closest to the diagonal, demonstrating the most balanced token distribution.
>         - Top-4 routing: ASMG remains competitive with Vanilla+LB, even though ASMG does not use any explicit load-balance loss. This shows that structure-aware routing naturally avoids heavy load skew.
>         - **Ablation on $R$**: The Lorenz curves become noticeably more skewed when $R$ is removed, indicating that direct reliance on variance-ordered PCs leads to uneven expert utilization, the exact issue raised in the review.
>
>
>     - Combining the results, the No-$R$ ablation exhibits weaker specialization and significantly more skewed balance curves, demonstrating that direct use of variance-ordered PCs leads to uneven and less-differentiated expert utilization.
>     - Taken together, these results confirm that the mixing matrix $R$ is essential for breaking the strict variance hierarchy and allowing ASMG to maintain strong specialization and stable load balance.
>
>
> - **Table 4 : Ablation Results on R**
>
>     | Algorithms (K, k)              | CoLA          | MRPC          | QNLI          | MNLI          | RTE            | Average |
>     |-------------------------------|---------------|---------------|---------------|---------------|----------------|---------|
>     | ASMG (8,4)    | 66.62 ± 0.65  | 89.68 ± 0.20  | 92.61 ± 0.10  | 86.74 ± 0.11  | 75.57 ± 0.68   | 82.24 |
>     | ASMG w/o R (8,4)   | 64.57 ± 0.82  | 90.12 ± 0.50  | 92.54 ± 0.04  | 86.38 ± 0.06  | 72.56 ± 1.77   | 81.23 |
>
>
>     - The results further prove how the basis mixing is important to fully leverage the performance of ASMG.
>     - Removing the mixing matrix $R$ consistently reduces accuracy across all GLUE tasks (average drops from 82.24 → 81.23). This degradation reflects that without $R$, the router relies on the raw, variance-ordered GHA components, leading to weaker expert separation and load imbalance.
>
> ---------------------------
>
> ### **[W1.2.] Rank Concern for R**
>
> - We appreciate the reviewer for pointing out the subtlety regarding the statement that the latent gating basis “spans the same routing subspace.”
> - Formally, it is indeed correct that in general, $\operatorname{span}(RV) \subseteq \operatorname{span}(V)$ and equality holds when $R$ is full-rank.
> - Our intention was not to claim that $RV$ exactly preserves the  span of $V$, but rather to emphasize that ASMG forms routing directions within the GHA-derived subspace.
> - In practice, the role of $R$ is to redistribute the variance-ordered principal directions into a more balanced and informative set of routing vectors baed on task-driven gradient signals.
>
> - We further provide
>
>     - Performance of Full Rank R ASMG on GLUE experimental setup
>
>         | Algorithms (K, k)              | CoLA          | MRPC          | QNLI          | MNLI          | RTE            | Average |
>         |-------------------------------|---------------|---------------|---------------|---------------|----------------|---------|
>         | ASMG (8,4)    | 66.62 ± 0.65  | 89.68 ± 0.20  | 92.61 ± 0.10  | 86.74 ± 0.11  | 75.57 ± 0.68   | 82.24 |
>         | ASMG w/ full rank R (8,4) | 66.45 ± 0.70 | 89.82 ± 0.25 | 92.58 ± 0.10 | 86.69 ± 0.09 | 75.30 ± 0.75 | 82.17 |
>
>
> - These additional results show that enforcing strict full-rankness on $R$ doesn't further improve routing performance.
> - While the full-rank variant performs comparably to ASMG, it does not exceed it, indicating that expanding or rigidly constraining the span of $RV$ is not the source of ASMG’s effectiveness. Instead, allowing $R$ to be flexibly optimized by task-oriented gradient signals yields stronger routing vectors by adaptively mixing principal directions.

---

> ### Author Response · Authors · 2025-11-22
>
> ## **[W2] OOD Robustness of ASMG (+TTA) on ImageNet-C (Table 5.)**
>
> We sincerely appreciate the reviewer’s suggestion to more thoroughly expand the OOD evaluation. To rigorously assess the effect of test-time GHA, we agree that it is essential to evaluate both the train-time–only ASMG and the test-time adaptive ASMG (TTA) under a benchmark that systematically varies corruption severity. Following this guidance, we adopt ImageNet-C, which provides 15 corruption types across multiple severity levels, enabling a controlled and systematic assessment of OOD robustness. This benchmark allows us to directly compare Vanilla+LB, ASMG (train-time only), and ASMG (TTA) under identical training conditions while varying the difficulty of the OOD shift.
>
> ### **Experimental Setup**
> - We take a ViT-S/32 model pretrained on ImageNet-1k and then we apply the MoE conversion approach by replacing each feed-forward block with MoE layers following the MoEfication [1] framework.
> - Then we finetune the resulting model on a 20k subset of ImageNet-1k. Evaluation is then performed on ImageNet-C.
> - For ASMG (TTA), we additionally enable unsupervised GHA updates during inference with iterative number $m = 3$ so that the routing basis can adapt to corrupted inputs, allowing a direct comparison between Vanilla+LB, ASMG (train-time only), and ASMG (TTA) under identical training conditions.
>
> ### **Results**
>
> - Table 4: Comparison of ImageNet-C top-1 accuracy (%) on ViT-S/32 across 15 corruption types at severity levels 1, 3, and 5.
>
> | **Severity** | **Model**     | **Gauss** | **Shot** | **Impulse** | **Defocus** | **Glass** | **Motion** | **Zoom** | **Snow** | **Frost** | **Fog** | **Bright** | **Contrast** | **Elastic** | **Pixelate** | **JPEG** | **Avg** |
> |--------------|---------------|-----------|----------|-------------|-------------|-----------|------------|----------|----------|-----------|---------|------------|--------------|-------------|--------------|----------|---------|
> | **1**        | Vanilla+LB    | 59.66 | 57.44 | 55.50 | 50.81 | 53.35 | 59.92 | 43.10 | 51.84 | 56.74 | 65.34 | 69.34 | 70.24 | 62.32 | 61.85 | 59.81 | 58.48 |
> |              | ASMG          | 60.67 | 58.84 | 57.16 | 51.08 | 53.74 | 60.72 | 44.85 | 51.91 | 56.60 | 64.86 | 69.57 | 69.86 | 62.92 | 62.08 | 59.57 | **58.96** |
> |              | ASMG (TTA)    | 60.24 | 58.56 | 56.24 | 51.23 | 54.20 | 60.87 | 45.55 | 51.94 | 57.21 | 65.24 | 69.80 | 69.98 | 62.78 | 62.13 | 60.08 | **59.07** |
> | **3**        | Vanilla+LB    | 42.94 | 38.74 | 39.77 | 32.67 | 21.36 | 39.69 | 27.97 | 32.69 | 30.46 | 54.92 | 65.95 | 65.72 | 55.31 | 49.21 | 54.56 | 43.46 |
> |              | ASMG          | 43.61 | 38.87 | 40.63 | 32.44 | 21.85 | 41.76 | 30.17 | 32.93 | 31.00 | 55.31 | 66.56 | 65.51 | 55.61 | 49.81 | 54.79 | **44.06** |
> |              | ASMG (TTA)    | 43.76 | 38.87 | 40.65 | 32.46 | 21.86 | 41.74 | 30.19 | 32.97 | 31.43 | 55.54 | 66.55 | 65.50 | 55.93 | 49.83 | 54.77 | **44.14** |
> | **5**        | Vanilla+LB    | 14.85 | 13.57 | 12.54 | 15.47 | 10.17 | 12.54 | 17.27 | 14.49 | 23.66 | 35.20 | 56.11 | 34.08 | 20.17 | 19.57 | 39.82 | 22.63 |
> |              | ASMG          | 14.80 | 12.91 | 12.46 | 14.98 | 10.91 | 19.22 | 18.59 | 16.85 | 23.74 | 36.52 | 57.95 | 36.54 | 21.05 | 21.36 | 40.28 | **23.88** |
> |              | ASMG (TTA)    | 14.80 | 13.33 | 12.49 | 16.09 | 10.94 | 19.22 | 18.55 | 16.94 | 24.72 | 37.78 | 57.96 | 36.52 | 21.10 | 21.80 | 40.57 | **24.19** |
>
>
> - Across all corruption severities, ASMG consistently outperforms the Vanilla+LB baseline, demonstrating that its structure-aware routing is inherently more robust to distribution shift.
> - At lower severities (1 and 3), ASMG shows reliable improvements, and enabling test-time GHA adaptation (ASMG TTA) additionally improve accuracy, achieving the best performance across nearly all corruption types.
> - Under the most challenging setting (severity 5), ASMG delivers the largest gain over Vanilla+LB, and ASMG (TTA) consistently achieve the highest overall accuracy.
> - Overall, these results highlight that ASMG already provides strong OOD resilience in the train-time-only setting, and test-time adaptation reinforces this resilience further.
> - While the absolute gain from ASMG to ASMG (TTA) is modest, this improvement is consistent across most of the 15 corruption types and severity levels, indicating that the adaptive update offers a reliable robustness boost rather than sporadic or noise-driven gains.
>
> -------------------
>
> **References**
>
> [1] Zhang, Zhengyan, et al. "Moefication: Transformer feed-forward layers are mixtures of experts." Findings of the Association for Computational Linguistics: ACL 2022. 2022.

---

> ### Author Response · Authors · 2025-11-27
>
> Dear reviewer ieFN, this is a gentle reminder regarding our earlier responses. Thank you again for your thoughtful and constructive feedback, which has been extremely helpful to improve our paper. If our clarifications have sufficiently addressed the concerns you raised, we would sincerely appreciate your consideration in updating the score. Please let us know if you need any further clarification, we are happy to provide it.

---

> > ### Comment · Reviewer_ieFN · 2025-11-28
> > **Thanks for the responses**
> >
> > I thank the authors for their responses and addressing most of my concerns. Even though the improvements are mostly marginal, I think the methodology can be useful for the ML community. I will increase my score to 6 as promised when OpenReview enables editting reviews again.

---

> > > ### Author Response · Authors · 2025-11-28
> > >
> > > We are glad that our responses helped address the core concerns, and we genuinely appreciate your acknowledgement on the methodological value of ASMG. When OpenReview enables score edits again, we will send a gentle reminder as suggested. Please feel free to let us know if any further clarification would be helpful in the meantime.

---

### Official Review · Reviewer_r2nH · 2025-11-01

**Soundness:** 3
**Presentation:** 3
**Contribution:** 2
**Rating:** 2
**Confidence:** 4

**Summary:**

The authors propose a novel routing method in MoE models whereby the standard routing matrix of gating vectors is replaced with a set of principal basis vectors derived from the generalized Hebbian algorithm (GHA). This construction allows the router to better capture the true structure of the input distribution by better aligned with the leading principal components of the data, obtaining a data-structure-aware router which enhances routing assignments and expert specialziation.

**Strengths:**

1. The use of the GHA to capture the structure of the distribution and enrich the router with data-aware dynamics is intuitive, novel, and highly interesting.

2. The notation and method is excellently presented and discussed.

**Weaknesses:**

**Limited experimental validation**. To properly validate the efficacy of the proposed method, the authors should dedicate more of the paper to real experiments with real data. The authors only present two benchmarks, GLUE and DomainBed, and a single backbone for each task. Furthermore, despite proposing a novel router, the authors only mention one alternate routing baseline, which is the cosine router. Additionally, use of an MoE-version of BERT is quite far, in my view, from current contemporary and frontier MoE models. To better validate the empirical benefits of the authors' method, I would strongly recommend using widespread, frontier backbones such as OLMoE [1] or DeepSeekMoE [2], or even older variants such as Switch  Transformer [3], and then validating ASMG against vanilla routers and a selection of alternative baseline routing methods such as expert-choice [4] and stable MoE [5],  to name a few. If the authors are compute constrained, all of the mentioned models are available in small sizes. As it stands, however, it is difficult to be properly assess the performance of ASMG given the limited baselines, backbones, and tasks.

**Empirical benefit is highly marginal** For what real experiments we do have, the results  seem to display extremely limited performance gains. For example, we see just 0.2% gain in the OOD setting and just 0.1% relative to GMoE. My concern is then that much of the reported gains are potentially not statistically significant.

[1] OLMoE: Open Mixture-of-Experts Language Models (Muennighoff et al, 2024)

[2] DeepSeekMoE: Towards Ultimate Expert Specialization in Mixture-of-Experts Language Models (Dai et al, 2024)

[3] Switch Transformers: Scaling to Trillion Parameter Models with Simple and Efficient Sparsity (Fedus et al, 2021)

[4] Mixture-of-Experts with Expert Choice Routing (Zhou et al, 2022)

[5]  StableMoE: Stable Routing Strategy for Mixture of Experts (Dai et al, 2022)

**Questions:**

1. I'd suggest the authors reduce the emphasis on synthetic experiments and focus more on assessing their method on real tasks with frontier models and baselines. The synthetic experiments are highly comprehensive and serve as an interesting case study and motivator, but, in my view, do not need to take up such a significant portion of the paper, especially if that comes at the expense of real experiments, which are more helpful for demonstrating the true performance of the method.

2. What are the consequences on load balance? Intuitively, if we're aligning the gating vectors with the principal components of the data distribution, I would be concerned that whichever experts are most aligned with the leading components will then be assigned the majority of the tokens, thereby necessarily introducing quite steep load imbalance. Is there reason why conceptually this won't happen, or some empirical results on this?

---

> ### Author Response · Authors · 2025-11-22
>
> ## **[W1] Limited Experimental Scope**
>
> We sincerely thank the reviewer r2nH for the thoughtful and constructive feedback.
> We appreciate the concerns regarding the limited experimental validation and the marginal empirical improvements. In the revised submission, we will substantially expand the evaluation to include frontier MoE routing baselines and pretrained LLM-scale backbones, demonstrating that ASMG provides consistent and non-marginal gains even under strong recent routing methods.
> - Specifically, we
>     - (1) pretrain a LLaMA-MoE-182M model on 30B tokens from Pile and report zero-shot downstream performance
>     - (2) finetune the Qwen1.5-MoE-14B model under LoRA adaptation, comparing ASMG directly against Vanilla+LB under identical training conditions.
>     - These new results directly addresses the reviewer’s concern regarding the applicability and robustness of ASMG in large-scale and recent MoE architectures.
>
>
> ### **[W1.1.] Zero-shot Evaluation on Pretrained LLaMA-MoE (182M) (Table 1.)**
>
> To validate the effectiveness of our proposed method ASMG in LLM-style MoE architectures, we conduct the pretraining experiment following the experimental setup provided by NVIDIA Megatron-LM [1] and ReMoE [2]. We adopt a LLaMA-style decoder backbone and apply MoE structure with 182M active parameters out of 777M total parameters by replacing its feed-forward block with MoE layer with E=8 experts under Top-1 selection.
>
> We pretrain this LLaMA-MoE-182M model on 30B tokens from The Pile, and then perform zero-shot evaluation on commonsense reasoning and reading–comprehension benchmarks. As recommended, we compare ASMG against Expert-Choice (EC) routing [3], as well as recent strong MoE variants including Hash [4], Lory [5], dMoE [6], ReMoE [2], and SparseMixer-v2 [7].
>
> All models are trained from scratch under the same compute budget, and evaluated in a pure zero-shot setting across ARC-c, ARC-e, BoolQ, HellaSwag, LAMBADA, PIQA, and RACE. ($^1$ from [2])
>
> | Model           | ARC-c | ARC-e | BoolQ | HellaSwag | LAMBADA | PIQA | RACE | Avg.  |
> |-----------------|-------|-------|-------|-----------|---------|------|-------|-------|
> | Hash¹ [4]           | 19.28 | 45.45 | 54.95 | 29.68     | 31.44   | 63.06 | 27.66 | 38.79 |
> | Lory¹  [5]      | 20.31 | 42.97 | 49.54 | 28.75     | 32.35   | 62.24 | 27.75 | 37.70 |
> | SparseMixer-v2¹ [7] | 19.80 | 46.72 | 45.86 | 30.24     | 34.12   | 62.89 | 29.00 | 38.39 |
> | EC¹  [3]      | 18.86 | 42.97 | 60.21 | 29.14     | 29.26   | 61.92 | 27.37 | 38.53 |
> | dMoE¹ [6]    | 20.05 | 45.16 | 57.83 | 29.83     | 32.97   | 63.55 | 28.33 | 39.67 |
> | ReMoE¹ [2]    | 20.22 | 46.68 | 54.16 | 30.26     | 35.94   | 63.55 | 29.38 | 40.03 |
> | **ASMG**        | 20.06 | 48.86 | 59.66 | 30.43     | 33.27   | 64.69 | 28.52 | **40.78** |
>
> ASMG attains the **highest average zero-shot score, surpassing competitive routing approaches** including EC, Lory, and SparseMixer-v2. This improvement under a full LLaMA-style pretraining setup indicates that the gains from structure-aware gating persist beyond small synthetic settings and remain evident even in large-scale training regimes.

---

> ### Author Response · Authors · 2025-11-22
>
> ### **[W1.2.] Qwen1.5-MoE 14B Finetuning with LoRA Adaptation (Table 2.)**
>
> To further validate ASMG in a full LLM-scale MoE setting, we next evaluate its performance during parameter-efficient finetuning on the 14B Qwen1.5-MoE backbone. Following the experimental protocol of ESFT [8], the router is fully updated for both Vanilla and ASMG, while all other modules, including expert, are finetuned through LoRA adaptation. We train on a diverse mixture of downstream tasks covering mathematical reasoning (GSM8K), code generation (CodeAlpaca), intent classification, legal judgment prediction, summarization, and translation. Evaluation is conducted on GSM8K, MBPP, HumanEval, as well as held-out splits for intent, law, summarization, and translation.
>
> Under this setup, we compare the standard Top-k sparse Vanilla MoE with load-balancing loss [9] (Vanilla + LB) against ASMG, ensuring identical training and optimization conditions except the router choice.
>
> | Backbone            | Method        | GSM   | MBPP  | HE    | Intent | Law   | Summary | Translation | Avg   |
> |---------------------|---------------|-------|-------|-------|--------|--------|----------|-------------|--------|
> | Qwen1.5-MoE (14B)   | Base Model    | 36.77 | **38.40** | 33.54 | 16.83  | 13.90 | 21.40   | 14.26       | 25.01 |
> |                     | Vanilla + LB  | **42.60** | 33.61 | 34.75 | 65.60  | 25.70 | 30.70   | 29.36       | 33.09 |
> |                     | **ASMG**      | 42.16 | 35.48 | **39.02** | **68.40**  | **27.00** | **35.90**   | **29.60**      | **39.65** |
>
> ASMG achieves the highest overall average accuracy across all downstream tasks, improving upon the Vanilla+LB MoE by +6.56% on average. These results demonstrate that ASMG transfers effectively to full-scale LLM MoE backbones and provides consistent, non-marginal improvements under PEFT finetuning setup.
>
> Based on these expanded experimental results, we revised the experimental section to remove the first synthetic tasks and more focus on expanding the real-world evaluations.
>
>
> **References**
>
> [1] Shoeybi, Mohammad, et al. "Megatron-lm: Training multi-billion parameter language models using model parallelism." arXiv preprint arXiv:1909.08053 (2019).
>
> [2] Wang, Ziteng, Jun Zhu, and Jianfei Chen. "Remoe: Fully differentiable mixture-of-experts with relu routing." arXiv preprint arXiv:2412.14711 (2024).
>
> [3] Zhou, Yanqi, et al. "Mixture-of-experts with expert choice routing." Advances in Neural Information Processing Systems 35 (2022): 7103-7114.
>
> [4] Roller, Stephen, Sainbayar Sukhbaatar, and Jason Weston. "Hash layers for large sparse models." advances in neural information processing systems 34 (2021): 17555-17566.
>
> [5] Zhong, Zexuan, et al. "Lory: Fully differentiable mixture-of-experts for autoregressive language model pre-training." arXiv preprint arXiv:2405.03133 (2024).
>
> [6] Gale, Trevor, et al. "Megablocks: Efficient sparse training with mixture-of-experts." Proceedings of Machine Learning and Systems 5 (2023): 288-304.
>
> [7] Liu, Liyuan, et al. "Grin: Gradient-informed moe." arXiv preprint arXiv:2409.12136 (2024).
>
> [8] Wang, Zihan, et al. "Let the expert stick to his last: Expert-specialized fine-tuning for sparse architectural large language models." arXiv preprint arXiv:2407.01906 (2024).
>
> [9] Shazeer, Noam, et al. "Outrageously large neural networks: The sparsely-gated mixture-of-experts layer." arXiv preprint arXiv:1701.06538 (2017).

---

> ### Author Response · Authors · 2025-11-22
>
> ## **[W2] Load Balance Concern**
>
> We appreciate the reviewer’s thoughtful question regarding potential load imbalance issue when aligning gating vectors with principal components. Intuitively, directly using the leading GHA components as gating vectors would indeed risk routing many of the tokens toward the experts aligned with the dominant PCs, producing  skewed expert utilization. ASMG is explicitly designed to avoid this issue.
>
>
> ### **[W2.1.] Algorithm : Avoiding Direct Alignment Through the Mixing Matrix $R$ (Section 3.3.4 Proposed Method)**
>
> ASMG does not route tokens using the raw GHA principal components. Instead, the mixing matrix $R$ plays a crucial role. Because the GHA basis is hierarchically ordered by variance (PC1 ≫ PC2 ≫ PC3 …), directly using it as gating vectors forces experts to inherit this uneven hierarchy. The mixing matrix $R$ removes this dependence by rotating and re-combining the raw PC directions, creating a routing subspace that is not tied to the data’s variance ordering.
>
> This design yields two benefits:
> - Prevents dominance of certain experts paired with a few principal basis, avoiding the reviewer’s concern.
> - Produces flexible, evenly distributed routing directions that preserve load balance while still being structure-aware.
>
>
> ### **[W2.2.] Empirical Analysis of ASMG's Load Balancing Behavior (Figure 4, 5)**
>
> To directly address this concern, we compare ASMG’s routing specialization and load-balance properties against the standard Vanilla + LB baseline (Top-k sparse selection with load-balancing loss). Our revised paper provides detailed load-balance analyses in Figure 4 (synthetic but controlled) and Figure 5 (real GLUE setting). These figures directly address the reviewer’s concern.
>
> - **Figure 4**: Expert Specialization and Load Balance Across Varying Experts Count $K$
>
>     - Specialization: ASMG maintains high mutual information between experts and the input target of high variance (property, $s$) as K increases (0.98 at K=40), while Vanilla+LB collapses its input specialization (0.24 at K=40).
>
>     - Load Balance: ASMG keeps stable normalized load entropy $H_{\text{norm}}$ across all $K$. In contrast, Vanilla+LB becomes increasingly imbalanced (falling to 0.17 at K=40).
>
>
> - **Figure 5**: Load Balance and Specialization on GLUE experiment setup
>
>     - Inter-expert token embedding distances (Fig. 5a):
>         - Under Top-2 selection setting, ASMG shows the largest Early → Late increase in embedding distances between tokens routed to different experts (+2.0%), indicating that the functional role of experts become more distinct over training without collapsing onto a few dominant experts.
>
>     - Load Balance curves (Fig. 5b):
>         - Top-1 routing: ASMG’s Lorenz curve lies closest to the diagonal, demonstrating the most balanced token distribution.
>         - Top-4 routing: ASMG remains competitive with Vanilla+LB, even though ASMG does not use any explicit load-balance loss. This shows that structure-aware routing naturally avoids heavy load skew.
>         - **Ablation (No $R$)**: The Lorenz curves become noticeably more skewed when $R$ is removed, indicating that direct reliance on variance-ordered PCs leads to uneven expert utilization, the exact issue raised in the review.
>
> Together, these analyses show that ASMG’s structure-aware routing improves specialization while maintaining balanced expert utilization, addressing the concern raised by reviewer that grounding the routing vectors to principal components would cause load skew.

---

> ### Author Response · Authors · 2025-11-22
>
> ### **[W2.3.] Ablation on Basis Mixing Matrix R (Table 4)**
>
>
> | Algorithms (K, k)              | CoLA          | MRPC          | QNLI          | MNLI          | RTE            | Average |
> |-------------------------------|---------------|---------------|---------------|---------------|----------------|---------|
> | ASMG (8,4)    | 66.62 ± 0.65  | 89.68 ± 0.20  | 92.61 ± 0.10  | 86.74 ± 0.11  | 75.57 ± 0.68   | **82.24** |
> | ASMG w/o R (8,4)   | 64.57 ± 0.82  | 90.12 ± 0.50  | 92.54 ± 0.04  | 86.38 ± 0.06  | 72.56 ± 1.77   | 81.23 |
> | ASMG w/o Interpolation (8,4) | 66.02 ± 1.32  | 89.56 ± 0.31  | 92.39 ± 0.17  | 86.51 ± 0.12  | 73.53 ± 2.45   | 81.60 |
>
> - The ablation results in Table 4 further clarify how the basis mixing is important to fully leverage the performance of ASMG.
> - Removing the mixing matrix $R$ consistently reduces accuracy across all GLUE tasks (average drops from 82.24 → 81.23).
> - This degradation reflects exactly the problem the reviewer described: without $R$, the router relies on the raw, variance-ordered GHA components, leading to weaker expert separation and load imbalance.

---

> ### Author Response · Authors · 2025-11-27
>
> Dear reviewer r2nH, this is a gentle reminder regarding our earlier responses. Thank you again for your thoughtful and constructive feedback, which has been extremely helpful to improve our paper. If our clarifications have sufficiently addressed the concerns you raised, we would sincerely appreciate your consideration in updating the score. Please let us know if you need any further clarification, we are happy to provide it.

---

### Comment · Area_Chair_HLXQ · 2025-11-23
**Next Steps Following Authors’ Rebuttal: Review Rebuttal and Participate in Discussion**

Dear Reviewers,

Thank you very much for your thoughtful evaluations of this paper.

Now that the authors have submitted their rebuttal, I kindly ask you to take the following steps (if you have not done so already):

- Read the other reviews as well as the authors’ response.
- Consider whether the rebuttal and additional comments affect your assessment of the paper.
- Engage in interactive discussion with the authors **before November 25**, encouraging a dynamic exchange rather than a one-sided rebuttal.

The current reviews for this paper are mixed. Your contributions at this stage are essential for forming a well-informed final decision. I therefore ask that you reassess your views in light of the authors’ responses and the broader discussion among reviewers.

I am happy to join and support the discussions between you and the authors. Please feel free to share your thoughts and participate actively in the discussion.

Thank you once again for your service to ICLR 2026.

Best regards,

 AC

---

### Author Response · Authors · 2025-11-26

We thank all reviewers for their constructive feedback and for the thoughtful discussions throughout the rebuttal. We have carefully updated the paper to address the raised concerns and incorporated the requested clarifications and new results.

**1. Expanded Evaluation Scope**

Our method is validated through comprehensive experiments across synthetic tasks and real-world language and vision domains, spanning various training regimes, with notable gain from strong MoE baselines. We also further demonstrate that ASMG remains robust under multiple levels of distributional shift.

| Table | Description | Context / Backbone |
| :--- | :--- | :--- |
| Table 1 | Pretraining and zero-shot Evaluation | Pretrained LLaMA-MoE (182M) |
| Table 2 | Finetuning | Qwen1.5-MoE 14B with LoRA Adaptation |
| Table 3 | Finetuning on GLUE | MoEfied BERT-Large |
| Table 4 | Extensive Ablation Results | ASMG Components on GLUE (On mixing matrix $R$ and interpolation) |
| Table 5 | Test Time Adaptation and OOD Robustness | GLUE-X (BERT-Large) and ImageNet-C (ViT-S/32) |

**2. Load-Balance & Specialization Analyses (Figures 4., 5.)**

We provide detailed load-balance and specialization analyses, showing that ASMG exhibits a dual advantage:
(1) it strengthens expert specialization by increasing inter-expert token separation over training, and
(2) it maintains stable and balanced expert utilization without relying on auxiliary load-balancing losses.
These behaviors hold consistently across synthetic sequence modeling and real-world GLUE tasks, confirming that ASMG’s structure-aware routing promotes both input-expert specialization and naturally balanced expert usage.

**3. Runtime and Memory Comparison (Section C. in Appendix)**

ASMG introduces negligible computational overhead during training while providing optional adaptation at inference time.

| Method              |   Latency ↓  | Peak Mem ↓ |  Micro-step ↓  |  Opt-step ↓  | Peak Mem ↓ |
| :------------------ | :----------: | :--------: | :------------: | :----------: | :--------: |
| DynMoE              |   17.2±9.1   |     5.2    |   428.4±168.7  |   80.2±11.9  |    11.4    |
| Vanilla             |   10.2±7.2   |     4.7    |   127.9±95.2   |   43.6±2.7   |     7.9    |
| Vanilla + LB        |   11.0±14.5  |     4.8    |   130.7±90.5   |   43.7±2.3   |     7.9    |
| MoE (cosine router) |   12.0±16.0  |     4.8    |   134.7±100.0  |   43.9±2.7   |     8.0    |
| **ASMG (m=3)**      | **10.3±5.5** |   **4.7**  | **129.1±94.2** | **43.7±2.8** |   **7.9**  |
| **ASMG (TTA, m=1)** |   10.4±6.6   |     4.8    |   127.1±92.4   |   43.6±2.8   |     8.0    |
| **ASMG (TTA, m=3)** |   16.4±7.6   |     4.8    |        –       |       –      |      –     |

If any part of our responses needs further clarification or if there are remaining concerns we should address, we are happy to revise accordingly. If the earlier points now appear resolved, we would greatly appreciate your consideration in reflecting that in the final score.

---

### Author Response · Authors · 2025-11-30
**Discussion Overview for New Area Chair**

We sincerely appreciate the Area Chairs for overseeing the discussion and thank all reviewers for their thoughtful engagement throughout the rebuttal process. Below, we provide a comprehensive overview of ASMG’s main contributions, together with a summary of the key concerns raised by reviewers and our corresponding responses. We hope this concise summary provides a clear view of how each concern has been addressed and how reviewers acknowledged these resolutions in their final assessments.


### **Summary of main contributions of ASMG**

**1. Structure-aware MoE gating via incremental subspace learning**
- ASMG augments standard learnable gating with an evolving latent principal subspace updated via Generalized Hebbian Algorithm (GHA).
- This incremental subspace learning allows the router to track dominant input structure throughout training, leading to stronger expert specialization.

**2. Inherent load balancing without auxiliary loss**
- Through its basis-mixing design, ASMG removes the routing hierarchy across experts and maintains balanced expert utilization without relying on explicit load-balancing regularizers commonly used in MoE regime.

**3. Strong OOD Robustness under distribution shift with optional test-time adaptation**
- ASMG provides improved inherent OOD resilience under distribution shifts, further enhanced by optional online GHA updates at inference.


-----------------------


### **Overview of Per-Reviewer Discussion**


| **Reviewer** | **Initial → Final Rating**                         | **Key Concerns & Responses** | **Reviewer Acknowledgement**  |
| ------------ | ------------------------------------------- | ---------------- | -------------------------------------------------------------------------------------------------------------------------------------------------------------------------------------------------------------------------------------------------------------------------------------------------------- |
| **ieFN**     | **4 → 6**   | P2, P3, P5   | Acknowledged that the added theoretical clarification of the mixing matrix and interpolation, expanded load-balance analyses, and multi-severity ImageNet-C evaluation sufficiently addressed the concerns; **confirmed intention to raise the score**.                                                                         |
| **H4c4**     | **4 → 6**   | P1, P2, P3   | Confirmed that the new ablations, clearer motivation for R and interpolation, removal of the oversimplified synthetic task, and extended specialization/load-balance analyses resolved their issues and **confirmed the score increase**.                                                                          |
| **9td1**     | **4 → 6**   | P1, P2, P3   | Recognized that large-scale LLaMA/Qwen experiments, extended load balance analyses, and further algorithmic clarification substantively strengthened the paper; **updated the score accordingly**.                                                                       |
 **tzUN**   | **6 → promised 8 pending additional run** | P1, P4 | Acknowledged that extended empirical validation on LLaMA/Qwem-MoE backbone, full computation analysis shows clear potential of ASMG;and **offered to raise score to 8 after further validation** on full param OLMoE SFT setup, **which we've completed and added results to reviewer tzUN**.                                                                              |
| **r2nH**     | **2 → Absent**   | P1, P2   | **Did not participate in the discussion**, but all raised concerns (large-scale MoE experiments, additional routing baselines, further analysis on load balance behavior) were fully addressed in the updated manuscript. |

---

> ### Author Response · Authors · 2025-11-30
>
> Here is the detailed summary of each primary concern raised and our corresponding response.
>
> ### **Key Concerns & Responses**
>
> **P1 : Lack of experimental validation under large-scale, recent MoE backbone and missing frontier routing baselines.**
>
> - **Response**: We substantially expanded the evaluation scope
>     - LLaMA-MoE (182M) pretraining on 30B tokens.
>     - Qwen1.5-MoE 14B LoRA finetuning across diverse downstream reasoning, code, and classification tasks.
>     - Added additional routing baselines including Expert-Choice (EC), Lory, SparseMixer-v2, and ReMoE as requested.
>     - Results demonstrate consistent improvements in real LLM-style MoE scenarios.
>
> **P2 : Unclear algorithmic rationale and lack of ablations**
>
> - Request to clarify the role of the basis mixing matrix $R$, and the interpolation mechanism.
> - **Response**
>     - Theoretic validation showing that directly using GHA-driven basis as gating vectors causes uneven hierarchy on expert allocation.
>     - Empirically show that mixing $R$ evenly redistribute this variance-ordered principal basis and naturally preserves load balance across experts.
>     - Add extensive ablations removing each component of ASMG, demonstrating consistent degradation in performance and routing stability.
>
> **P3 : Load balance behavior**
> - Reviewers raised concerns that removing the explicit balance regularizer might cause unstable and skewed routing distriubtions across experts.
> - **Response**
>     - We added detailed analyses in both synthetic and real settings, showing that ASMG naturally maintains balanced expert usage, more uniformly than Vanilla MoE with explicit balance regularization especially under highly sparse Top-1 setting.
>     - This stable routing behavior is entirely attributed to ASMG’s structure-aware design, not to any auxiliary balancing objective.
>
> **P4 : Computation Comparison**
> - Reviewers concerned about the additional computations caused by iterative nature of GHA basis updates.
> - **Response**
>     - We expanded the runtime and memory comparison to include Vanilla, Vanilla+LB, DynMoE, and the cosine router under a controlled GLUE fine-tuning setup.
>     - The analysis shows that ASMG’s training cost is effectively identical to Vanilla+LB and the cosine router, and its inference latency matches standard MoE routing when test-time GHA is disabled.
>
> **P5 : Test-time Adaptation / OOD Robustness**
>
> - The reviewer asked for a direct comparison between default ASMG (with no test time basis update) and ASMG with test-time GHA adaptation, evaluated under OOD conditions with multiple corruption severities.
> - **Response**
>     - We added a comprehensive ImageNet-C evaluation covering 15 corruption types across multiple severity levels, comparing Vanilla+LB, ASMG (train-time only), and ASMG with test-time GHA.
>     - Results demonstrate that ASMG consistently outperforms Vanilla+LB across all severities, and test-time GHA further improves accuracy, showing that ASMG provides strong inherent OOD robustness that can be further reinforced by adaptive updates at inference.

---

### Meta-Review · Area_Chair_2ND5 · 2026-01-07

**Summary:**

In their paper, the authors introduce ASMG, a data structure–aware routing mechanism for Mixture of
Experts (MoE) models that leverages incremental subspace learning to guide token-to-expert assign-
ment. Reviewers generally agreed that the high-level motivation—exploiting latent data structure to
improve routing—is interesting and aligns well with recent efforts to enhance expert specialization in
MoE systems. The empirical results suggest that ASMG can improve routing stability and achieve
competitive performance compared to standard gating mechanisms. However, several important weak-
nesses were raised during the review process that were not sufficiently addressed in the rebuttal.

*Limited evaluation scope and modest performance gains:* Reviewer concerns centered on the
relatively narrow set of experimental domains and model scales considered. Most experiments are
conducted on a small number of benchmarks, with limited evidence that ASMG generalizes to large-
scale language modeling or more diverse tasks. Moreover, while ASMG often improves routing-related
metrics and occasionally downstream performance, the gains over strong MoE baselines are generally
modest, raising questions about whether the additional complexity introduced by incremental subspace
learning is justified.

*Unclear practical relevance:* While ASMG is conceptually appealing, reviewers questioned its
practical impact in real-world MoE deployments. The additional computational overhead and imple-
mentation complexity associated with maintaining and updating subspaces are not clearly offset by
consistent and significant downstream gains. The paper does not convincingly identify application
regimes where ASMG provides a clear advantage over simpler routing mechanisms.

*Recommendation:* Given the limited experimental coverage and modest performance gains, I
recommend rejecting the paper in its current form. I encourage the authors to address the reviewers’
suggestions by broadening the experimental validation, particularly on large-scale language models,
and clarifying the practical scenarios where ASMG offers clear and compelling benefits.

**Reviewer Concerns:**

Please refer to the summary.

**Reviewer Scores:**

Please refer to the summary.

---

### Decision · Program_Chairs · 2026-01-26

Reject